# AP: Selective Activation for De-sparsifying Pruned Networks

**Shiyu Liu**                                                           *shiyu_liu@u.nus.edu*
*Department of Electrical and Computer Engineering*
*College of Design and Engineering*
*National University of Singapore*

**Rohan Ghosh**                                                        *rghosh92@gmail.com*
*Department of Electrical and Computer Engineering*
*College of Design and Engineering*
*National University of Singapore*

**Mehul Motani**                                                       *motani@nus.edu.sg*
*Department of Electrical and Computer Engineering*
*College of Design and Engineering*
*N.1 Institute for Health*
*Institute of Data Science*
*Institute for Digital Medicine (WisDM)*
*National University of Singapore*

**Reviewed on OpenReview:** *https://openreview.net/forum?id=EGQSpkUDdD*

## Abstract

The rectified linear unit (ReLU) is a highly successful activation function in neural networks as it allows networks to easily obtain sparse representations, which reduces overfitting in overparameterized networks. However, in the context of network pruning, we find that the sparsity introduced by ReLU, which we quantify by a term called dynamic dead neuron rate (DNR), is not beneficial for the pruned network. Interestingly, the more the network is pruned, the smaller the dynamic DNR becomes after optimization. This motivates us to propose a method to explicitly reduce the dynamic DNR for the pruned network, i.e., de-sparsify the network. We refer to our method as Activate-while-Pruning (AP). We note that AP does not function as a stand-alone method, as it does not evaluate the importance of weights. Instead, it works in tandem with existing pruning methods and aims to improve their performance by selective activation of nodes to reduce the dynamic DNR. We conduct extensive experiments using various popular networks (e.g., ResNet, VGG, DenseNet, MobileNet) via two classical and three competitive pruning methods. The experimental results on public datasets (e.g., CIFAR-10, CIFAR-100) suggest that AP works well with existing pruning methods and improves the performance by 3% - 4%. For larger scale datasets (e.g., ImageNet) and competitive networks (e.g., vision transformer), we observe an improvement of 2% - 3% with AP as opposed to without. Lastly, we conduct an ablation study and a substitution study to examine the effectiveness of the components comprising AP.

## 1 Introduction

The rectified linear unit (ReLU) (Glorot et al., 2011), $\sigma(x) = \max\{x, 0\}$, is the most widely used activation function in neural networks (e.g., ResNet (He et al., 2016), Transformer (Vaswani et al., 2017)). The success of ReLU is mainly due to fact that existing networks tend to be overparameterized and ReLU can easily regularize overparameterized networks by introducing sparsity (i.e., post-activation output is zero) (Li et al., 2023; Denil et al., 2014; Wang et al., 2023), leading to promising results in many computer vision tasks (e.g., image classification (He et al., 2016), object detection (Dai et al., 2021; Joseph et al., 2021)).

In this paper, we study the ReLU's sparsity constraint in the context of network pruning (i.e., a method of compression that removes weights from the network). Specifically, we question the utility of ReLU's sparsity constraint, when the network is no longer overparameterized during iterative pruning. In the following, we summarize the workflow of our study together with our contributions.

1. **Motivation and Theoretical Study.** In Section 3.1, we introduce a term called dynamic Dead Neuron Rate (DNR), which quantifies the sparsity introduced by ReLU neurons that are not completely pruned during iterative pruning. Through rigorous experiments on existing networks (e.g., ResNet (He et al., 2016)), we find that the more the network is pruned, the smaller the dynamic DNR becomes during and after optimization. This suggests that the sparsity introduced by ReLU is not beneficial for pruned networks. Further theoretical investigations also reveal the importance of reducing dynamic DNR for pruned networks from an information bottleneck (IB) (Tishby & Zaslavsky, 2015) perspective (see Section 3.2).

2. **A Method for De-sparsifying Pruned Networks.** In Section 3.3, we propose a method called Activate-while-Pruning (AP) which aims to explicitly reduce dynamic DNR. We note that AP does not function as a stand-alone method, as it does not evaluate the importance of weights. Instead, it works in tandem with existing pruning methods and aims to improve their performance by reducing dynamic DNR. AP has two variants: (i) AP-Lite which slightly improves the performance of existing methods, but without increasing the algorithm complexity, and (ii) AP-Pro which introduces an additional retraining step to the existing methods in every pruning cycle, but significantly improves the performance of existing methods.

3. **Experiments.** In Section 4, we conduct experiments on CIFAR-10, CIFAR-100 (Krizhevsky et al., 2009) with various popular networks (e.g., ResNet, VGG (Simonyan & Zisserman, 2014), MobileNet (Sandler et al., 2018), DenseNet (Huang et al., 2017)) using two classical and three competitive pruning methods. The results demonstrate that AP works well with existing pruning methods and improve their performance by 3% - 4%. For the larger scale dataset (e.g., ImageNet (Deng et al., 2009)) and competitive networks (e.g., vision transformer (Dosovitskiy et al., 2020)), we observe an improvement of 2% - 3% with AP as opposed to without.

4. **Ablation Study.** In Section 4.3, we carry out an ablation study to further investigate and demonstrate the effectiveness of several key components that make up the proposed AP.

5. **Substitution Study.** In Section 4.4, we conduct a substitution study to replace certain components in the proposed AP and examine the impact on pruning performance.

## 2 Background

Network pruning is a method used to reduce the size of the neural network, with its first work (LeCun et al., 1998) dating back to 1990. In terms of the pruning style, all existing methods can be divided into two classes: (i) **One-Shot Pruning** and (ii) **Iterative Pruning**. Assuming that we plan to prune $Q\%$ of the parameters of a trained network, a typical **pruning cycle** consists of three basic steps: (i) Prune $\eta\%$ of existing parameters based on given metrics (ii) Freeze pruned weights as zero (iii) Retrain the pruned network to recover the performance. In One-Shot Pruning, $\eta$ is set to $Q$ and the parameters are pruned in one pruning cycle. While for Iterative Pruning, a much smaller portion of parameters (i.e., $\eta << Q$) are pruned per pruning cycle. The pruning process is repeated multiple times until $Q\%$ of parameters are pruned. As for performance, Iterative Pruning often results in better performance compared to One-Shot Pruning (Li et al., 2017; Vysogorets & Kempe, 2023; Zhang & Freris, 2023). So far, existing works aim to improve the pruning performance by exploring either new pruning metrics or new retraining methods.

**Pruning Metrics.** Weight magnitude is the most popular approximation metric used to determine less useful connections; the intuition being that smaller magnitude weights have a smaller effect in the output, and hence are less likely to have an impact on the model outcome if pruned (He et al., 2020; Li et al., 2020a;b). Many works have investigated the use of weight magnitude as the pruning metric (Han et al., 2015; Frankle & Carbin, 2019). More recently, Lee et al. (2020) introduced layer-adaptive magnitude-based pruning (LAMP) and attempted to prune weights based on a scaled version of the magnitude. Park et al. (2020) proposed a method called Lookahead Pruning (LAP), which evaluates the importance of weights based on the impact of pruning on neighbor layers. Another popular metric used for pruning is via the gradient, the intuition

being that weights with smaller gradients are less impactful in optimizing the loss function. Examples are (LeCun et al., 1998; Theis et al., 2018), where LeCun et al. (1998) proposed using the second derivative of the loss function with respect to the parameters (i.e., the Hessian Matrix) as a pruning metric and Theis et al. (2018) used Fisher information to approximate the Hessian Matrix. A recent work (Blalock et al., 2020) reviewed numerous pruning methods and suggested two classical pruning methods for performance evaluation: (i) **Global Magnitude**: Pruning the weights with the lowest absolute value anywhere in the network and (ii) **Global Taylor** (Molchanov et al., 2019): Pruning the weights with the lowest absolute value of (weight × gradient) anywhere in the network.

1. **Global Magnitude**: Pruning weights with the lowest absolute value globally (anywhere in the network).

2. **Global Taylor**: Pruning weights with the lowest absolute value of (weight×gradient) globally.

**Retraining Methods.** Another factor that significantly affects the pruning performance is the retraining method. For example, Han et al. (2015) trained the unpruned network with a learning rate (LR) schedule and retrained the pruned network using a constant learning rate (i.e., often the final LR of the LR schedule). A recent work (Renda et al., 2019) proposed learning rate rewinding which used the same learning rate schedule to retrain the pruned network, leading to a better pruning performance. More recently, Liu et al. (2021a) attempted to optimize the choice of LR during retraining and proposed a LR schedule called S-Cyc. They showed that S-Cyc could work well with various pruning methods, further improving the existing performance. Most notably, Frankle & Carbin (2019) found that resetting the unpruned weights to their original values (known as **weight rewinding**) after each pruning cycle could lead to even higher performance than the original model. Some follow-on works (Zhou et al., 2019; Renda et al., 2019; Malach et al., 2020; Evci et al., 2021) investigated this phenomenon more precisely and applied this method in other fields (e.g., transfer learning (Mehta, 2019), reinforcement learning and natural language processing (Yu et al., 2020)) while other works (Evci et al., 2022; Paul et al., 2022) study its limitation and attempt to improve on it. One interesting work to mention is (Chen et al., 2022), which further examined the lottery ticket hypothesis from other perspectives, such as interpretability and geometry of loss landscapes.

**Other Works.** In addition to works mentioned above, several other works also share some deeper insights on network pruning (Liu et al., 2019; Zhu & Gupta, 2018; Li et al., 2022; Wang et al., 2022; Peste et al., 2021; Lee & et al, 2023; Gale et al., 2019). For example, Wang et al. (2020) suggested that the fully-trained network could reduce the search space for the pruned structure. More recently, Luo & Wu (2020) addressed the issue of pruning residual connections with limited data and Ye et al. (2020) theoretically proved the existence of small subnetworks with lower loss than the unpruned network. You et al. (2022) motivated the use of the affine spline formulation to analyze recent pruning techniques. Liu et al. (2022a) applied the network pruning technique in graph networks and approximated the subgraph edit distance.

## 3 Activate-while-Pruning

In Section 3.1, we first conduct experiments to evaluate the DNR during iterative pruning. Next, in Section 3.2, we link the experimental results to theoretical studies and motivate Activate-while-Pruning (AP). In Section 3.3, we introduce the idea of AP and present its algorithm. Lastly, in Section 3.4, we illustrate how AP can improve on the performance of existing pruning methods.

### 3.1 Experiments on DNR

We study the state of the ReLU function during iterative pruning and introduce a term called Dead Neuron Rate (DNR), which is the percentage of dead ReLU neurons (i.e., a neuron with a post-ReLU output of zero) in the network averaged over all training samples when the network converges. Mathematically, the DNR can be written as

$$\text{DNR} = \frac{1}{n} \sum_{i=1}^{n} \frac{\text{\# of dead ReLU neurons}}{\text{all ReLU neurons in the unpruned network}} \tag{1}$$

$$= \frac{1}{n} \sum_{i=1}^{n} \frac{\text{\# of dynamically + statically dead ReLU neurons}}{\text{all ReLU neurons in the unpruned network}} \tag{2}$$

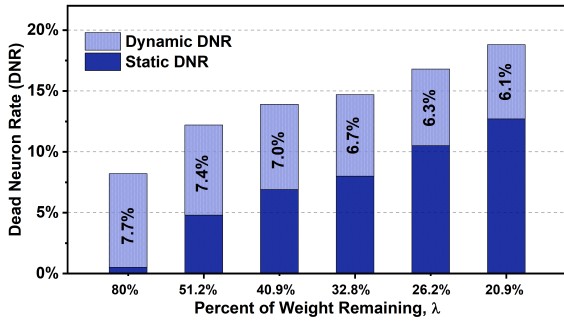 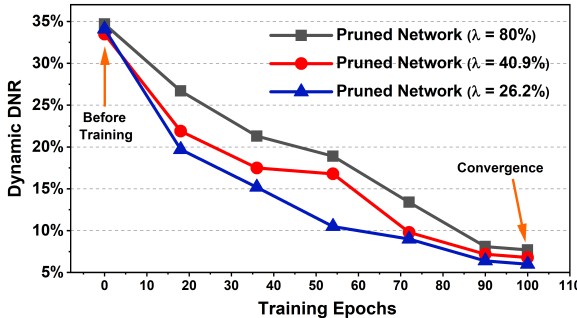

Figure 1: **Left**: Dynamic and static Dead Neuron Rate (DNR) when iteratively pruning ResNet-20 using Global Magnitude; **Right**: The corresponding dynamic DNR during optimization.

where $n$ is the number of training samples. We further classify a dead neuron as either dynamically dead or statically dead. The **dynamically dead neuron** is a dead neuron in which not all of the weights have been pruned. Hence, it is not likely to be permanently dead and its state depends on its input. As an example, a neuron can be dead for a sample $X$, but it could be active (i.e., post-ReLU output $> 0$) for a sample $Y$. The DNR contributed by dynamically dead neurons is referred to as **dynamic DNR**. The **statically dead neuron** is a dead neuron in which all associated weights have been pruned. The DNR contributed by statically dead neurons is referred to as **static DNR**.

**Related Works.** The phenomenon of dead ReLU neurons is a widely studied topic and inspire many interesting works Sokar et al. (2023); Dohare et al. (2021). DNR is a term that we introduce to quantify the sparsity introduced by ReLU. We note that DNR is closely related to activation sparsity (Kurtz & et al, 2020; Georgiadis, 2019) in the literature and in many scenarios, they can be interchangeable. The only difference could be that DNR is a more explicit term which quantifies the percent of dead ReLU neurons in the network. In the literature, many similar sparsity metrics have also been proposed (Hurley & Rickard, 2009). As an example, the Gini Index (Goswami et al., 2016) computed from Lorenz curve (i.e., plot the cumulative percentages of total quantities) can be used to evaluate the sparsity of network graphs. Another popular metric will be Hoyer measure (Hoyer, 2004) which is the ratio between L1 and L2 norms, can also be used to evaluate the sparsity of networks. Another interesting metric to mention will be parameter sparsity (Goodfellow et al., 2016) which computes the percentage of zero-magnitude parameters among all parameters. Both parameter sparsity and DNR will contribute to sparse representations, and in this paper, we use DNR to quantify the sparsity introduced by ReLU.

**Experiment Setup and Observations.** Given the definition of DNR, static and dynamic DNR, we conduct pruning experiments using ResNet-20 on the CIFAR-10 dataset with the aim of examining the benefit (or lack thereof) of ReLU's sparsity for pruned networks. We iteratively prune ResNet-20 with a pruning rate of 20 (i.e., 20% of existing weights are pruned) using the Global Magnitude (i.e., prune weights with the smallest magnitude anywhere in the network). We refer to the standard implementation reported in (Renda et al., 2019) (i.e., SGD optimizer (Ruder, 2016), 100 training epochs and batch size of 128, learning rate warmup to 0.03 and drop by a factor of 10 at 55 and 70 epochs, with learning rate rewinding (Renda et al., 2019), but without weight rewinding (Frankle & Carbin, 2019)) and compute the static DNR and dynamic DNR while the network is iteratively pruned. The experimental results are shown in Fig. 1, where we make two observations as follows.

1. As shown in Fig. 1 (left), the value of DNR (i.e., sum of static and dynamic DNR) increases as the network is iteratively pruned. As expected, static DNR grows as more weights are pruned.

2. Surprisingly, dynamic DNR tends to decrease as the network is iteratively pruned (see Fig. 1 (left)), suggesting that pruned networks do not favor the sparsity of ReLU. In Fig. 1 (right), for pruned networks with different $\lambda$ (i.e., percent of remaining weights), they have similar dynamic DNR at beginning, but the pruned network with smaller $\lambda$ tends to have a smaller dynamic DNR during and after optimization.

**Result Analysis.** One possible reason for the decrease in dynamic DNR could be due to the fact that once the neuron is dead, its gradient becomes zero, meaning that it stops learning and degrades the learning ability of the network (Lu et al., 2019; Arnekvist et al., 2020). This could be beneficial as existing networks tend to be overparameterized and dynamic DNR may help to reduce the occurrence of overfitting. However, for pruned networks whose learning ability are heavily degraded, the dynamic DNR could be harmful as a dead ReLU always outputs the same value (zero as it happens) for any given non-positive input, meaning that it takes no role in discriminating between inputs. Therefore, during retraining, the pruned network attempts to restore its performance by reducing its dynamic DNR so that the extracted information can be passed to the subsequent layers. Similar performance trends can be observed using VGG-19 with Global Taylor (see Fig. 4 in the **Appendix**). Next, we present a **theoretical study** of DNR and show its relevance to the network's ability to discriminate.

### 3.2 Theoretical Insights: Relevance to Information Bottleneck and Complexity

Here, we present some theoretical results and subsequent insights that highlight the relevance of the dynamic DNR of a certain layer of the pruned network to the Information Bottleneck (IB) method proposed in (Tishby & Zaslavsky, 2015). In the IB setting, the computational flow is denoted as $X \to T \to Y$, where $X$ represents the input, $T$ represents the extracted representation, and $Y$ represents the network's output. In (Tishby & Zaslavsky, 2015), the authors observed that the training of neural networks is essentially a process of minimizing the mutual information (Cover & Thomas, 2006) between $X$ and $T$ (denoted as $I(X;T)$) while keeping $I(Y;T)$ large (precisely what IB suggests). A consequence of this is that over-compressed features (very low $I(X;T)$) will not retain enough information to predict the labels, whereas under-compressed features (high $I(X;T)$) imply that more label-irrelevant information is retained in $T$ which can adversely affect generalization performance. Next, we provide a few definitions.

**Definition 1. Layer-Specific dynamic DNR ($D_{DNR}(T)$):** We are given a dataset $S = \{X_1, ..., X_m\}$, where $X_i \sim P \; \forall i$ (i.i.d) and $P$ is the data generating distribution. We denote the dynamic DNR of the neurons at a certain layer within the network represented by the vector $T$, by $D_{DNR}(T)$. $D_{DNR}(T)$ is computed over the entire distribution of input in $P$.

**Definition 2. Layer-Specific static DNR ($S_{DNR}(T)$):** In the same manner as $D_{DNR}(T)$, we define the layer-specific static DNR of a network layer $T$.

With this, we now outline our first theoretical result which highlights the relevance of $D_{DNR}(T)$ and $S_{DNR}(T)$ to $I(X;T)$, as follows.

**Theorem 1.** We are given the computational flow $X \to T \to Y$, where $T$ represents the features at some arbitrary layer within a network, which are represented with finite precision (e.g., float32 or float64). We consider the subset of network configurations for which (a) the activations in $T$ are less than a threshold $\tau$ and (b) the zero-activation probability of each neuron in $T$ is upper bounded by some $p_S < 1$. Let $dim(T)$ represent the dimensionality of $T$, i.e., the number of neurons at that depth. We then have,

$$I(X;T) \leq C \times dim(T) \times \left(1 - S_{DNR}(T) - D_{DNR}(T)\left(1 - \frac{1}{C}\log\frac{1 - S_{DNR}(T)}{D_{DNR}(T)}\right)\right), \tag{3}$$

for a finite, independent constant $C$ that only depends on the network architecture, $\tau$ and $p_S$.

The following corollary addresses the dependencies of Theorem 1. The proof of Theorem 1 and Corollary 1 are provided in the **Appendix**.

**Corollary 1.** In the setting of Theorem 1, the right hand side of equation 3 decreases as $D_{DNR}(T)$ or $S_{DNR}(T)$ increases.

**Remark 1. (Relevance to Complexity)** We see that (Shamir et al., 2010) notes how the metric $I(X;T)$ represents the *effective complexity* of the network. As Theorem 3 in (Shamir et al., 2010) shows, $I(X;T)$ captures the dependencies between $X$ and $T$ and directly correlates with the network's ability to fit the data. Coupled with the observations from Theorem 1 and Corollary 1, for a fixed pruned network configuration

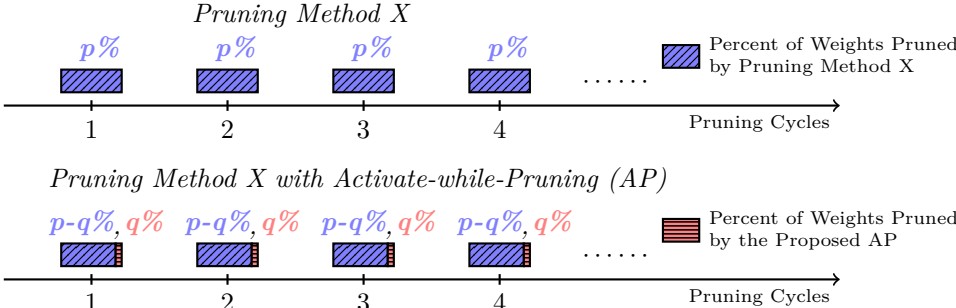

Figure 2: Illustration of how AP works in tandem with existing pruning methods (e.g., method X).

(i.e., fixed $S_{DNR}(T)$), greater $D_{DNR}(T)$ will likely reduce the *effective complexity* of the network, undermining the function-fitting ability of the neural network.

**Remark 2. (Motivation for AP)** Theorem 1 also shows that a pruned network, which possesses large $S_{DNR}(T)$, leads to a higher risk of *over-compression* of information (low $I(X;T)$). To address this issue, we can reduce the dynamic DNR (from Corollary 1) so that the upper bound of $I(X;T)$ can be increased, mitigating the issue of *over-compression* for a pruned network. This agrees with our initial motivation that the sparsity introduced by ReLU is not beneficial for the pruned network and reducing dynamic DNR helps in avoiding over-compressed features while simultaneously increasing the effective complexity of the network.

### 3.3 Algorithm of Activate-while-Pruning

The experimental and theoretical results above suggest that, in order to better preserve the learning ability of pruned networks, a smaller dynamic DNR is preferred. This motivates us to propose Activate-while-Pruning (AP) which aims to explicitly reduce dynamic DNR.

We note that the proposed AP does not work alone, as it does not evaluate the importance of weights. Instead, it serves as a booster to existing pruning methods and help to improve their pruning performance by reducing dynamic DNR (see Fig. 2). Assume that the pruning method X removes $p\%$ of weights in every pruning cycle (see the upper part in Fig. 2). After using AP, the overall pruning rate remains unchanged as $p\%$, but $(p-q)\%$ of weights are pruned according to the pruning method X with the aim of pruning less important weights, while $q\%$ of weights are pruned according to AP (see the lower part in Fig. 2) with the aim of reducing dynamic DNR. Consider a network $f(\theta)$ with ReLU activation function. Two key steps to reducing dynamic DNR are summarized as follows.

**(1) Locate Dead ReLU Neurons.** Consider a neuron in the hidden layer with ReLU activation function, taking $n$ inputs $\{X_1W_1, ..., X_nW_n | X_i \in \mathbb{R}$ is the input and $W_i \in \mathbb{R}$ is the associated weight$\}$. Let $j$ be the pre-activated output of the neuron (i.e., $j = \sum_{i=1}^{n} X_i W_i$) and $\mathcal{J}$ be the post-activated output of the neuron ($\mathcal{J} = ReLU(j)$). Let $\mathcal{L}$ be the loss function and assume the neuron is dead ($\mathcal{J} = 0$), then the gradient of its associated weights (e.g., $W_1$) with respected to the loss function will be $\frac{\partial \mathcal{L}}{\partial W_1} = \frac{\partial \mathcal{L}}{\partial \mathcal{J}} \cdot \frac{\partial \mathcal{J}}{\partial j} \cdot \frac{\partial j}{\partial W_1} = 0$ as $\frac{\partial \mathcal{J}}{\partial j} = 0$. If a neuron is often dead during training, the weight movement of its associated weights is likely to be smaller than other neurons. Therefore, we compute the difference between weights at initialization ($\theta_0$) and the weights when the network convergences ($\theta_*$), i.e., $|\theta_* - \theta_0|$ and use it as a heuristic to locate dead ReLU neurons.

**(2) Activate Dead ReLU Neurons.** Assume we have located a dead neuron in the hidden layer with $n$ inputs $\{X_1W_1, ..., X_nW_n | X_i \in \mathbb{R}$ is the input and $W_i \in \mathbb{R}$ is the associated weight$\}$. We note that $X_i$ is non-negative as $X_i$ is usually the post-activated output from the previous layer (i.e., the output of ReLU is non-negative). Therefore, a straightforward way to activate the dead neuron is to prune the weights with the negative value. By pruning such negative weights, we can increase the value of the pre-activation output, which may turn the pre-activation output into positive so as to reduce dynamic DNR.

---

**Algorithm 1** The Pruning Metric of the Proposed AP

---

**Require:** (i) Network $f$ with unpruned weights $\theta_0$ at initialization, $f(\theta_0)$; (ii) Network $f$ with unpruned weights $\theta_*$ at convergence, $f(\theta_*)$; (iii) Pruning Rate of AP, $q$;
    **Locate Dead Neurons**: Sort $|\theta_* - \theta_0|$ in an ascending order.
    **Activate Dead Neurons**: In the ascending order of $|\theta_* - \theta_0|$, prune first $q\%$ negative weights.

---

---

**Algorithm 2** The Pruning Method X with and without AP

---

**Require:** (i) Network, $f(\theta)$; (ii) Pruning Rate of Method X, $p$; (iii) Pruning Rate of AP, $q$; (iv) Pruning Cycles, $n$; (v) Pro_Flag = {0: AP-Lite, 1: AP-Pro};
———————————————— **The Conventional Pruning Method X** ————————————————
    **for** $i = 1$ to $n$ **do**
        Randomly initialize unpruned weights, $\theta \leftarrow \theta_0$.
        Train the network to convergence, arriving at parameters $\theta_*$.
        Prune $p$ % of $\theta_*$ according to the pruning method X.
    **end for**
    Retrain: Retrain the network to recover its performance.
———————————— **The Conventional Pruning Method X with Proposed AP** ————————————
    **for** $i = 1$ to $n$ **do**
        Randomly initialize unpruned weights, $\theta \leftarrow \theta_0$.
        Train the network to convergence, arriving at parameters $\theta_*$.
        Prune $(p - q)$ % of $\theta_*$ according to the pruning method X.
        **if** Pro_Flag **then**                     # Execution of AP-Pro
            *(i) Pruning: Prune $q$ % of parameter $\theta_*$ according to the metric of AP (see details in Algo. 1).*
            *(ii) Weight Rewinding: Reset the remaining parameters to their values in $\theta_0$.*
            *(iii) Retrain: Retrain the pruned network to recover its performance.*
        **end if**
    **end for**
    **if** NOT Pro_Flag **then**                 # Execution of AP-Lite
        *(i) Pruning: Prune $q$ % of the parameters $\theta_*$ according to the metric of AP (see details in Algo. 1).*
        *(ii) Weight Rewinding: Reset the remaining parameters to their values in $\theta_0$.*
        *(iii) Retrain: Retrain the pruned network to recover its performance.*
    **end if**

---

## 3.4 How AP Improves Existing Methods

We now summarize how AP can improve existing pruning methods in Algorithm 2, where the upper part is the algorithm of a standard iterative pruning method called pruning method X and the lower part is the algorithm of method X with AP. The proposed AP has two variants: **AP-Pro** and **AP-Lite**. We note that both AP-Pro and AP-Lite contain the same three steps, summarized as follows.

- **Step 1: Pruning.** Given a network at convergence with a set of dynamically dead ReLU neurons, $\mathcal{N}_1 = \{n_1, n_2, n_3, ...\}$. The pruning step of AP aims to activate these dynamically dead ReLU neurons by pruning negative weights (i.e., see Algorithm 1), so as to preserve the learning ability of the pruned network.

- **Step 2: Weight Rewinding.** Resetting unpruned weights to their values at the initialization. We note that different weight initializations could lead to different sets of $\mathcal{N}$. In step 1, AP aims to reduce dynamic DNR for the target $\mathcal{N}_1$ and weight rewinding attempts to prevent the target $\mathcal{N}_1$ from changing too much. Since the weights of ReLU neurons in $\mathcal{N}_1$ have been pruned by AP, these neurons could become active during retraining. The effect of weight rewinding is evaluated via an ablation study.

- **Step 3: Retraining.** Retrain the pruned network to recover performance.

**AP-Lite and AP-Pro.** The key difference between AP-Lite and AP-Pro is that AP-Lite applies these three steps only once at the end of pruning. It aims to slightly improve the performance, but does not substantially increase the algorithm complexity. For AP-Pro, it applies the three steps above in every pruning cycle, which

increases the algorithm complexity (mainly due to the retraining step), but aims to significantly improve the performance, which could be preferred in performance oriented tasks.

**Difference with Existing Works.** The AP is a pruning method which does not work alone as AP's pruning metric cannot evaluate the importance of weights (verified in Section 4.3). AP works in tandem with existing pruning methods and help to further pruned the network by reducing the occurrence of dynamic dead neurons (i.e., decrease activation sparsity). This is different from existing works (Raihan & Aamodt, 2020; Liu et al., 2022b; Akiva-Hochman et al., 2022; Gupta et al., 2019) which jointly optimize weight and activation sparsity for computation acceleration, the proposed AP investigates the interaction of weight and activation sparsity from a new perspective, i.e., how to tradeoff activation sparsity for more weight sparsity.

## 4 Performance Evaluation

In Section 4.1, we first summarize the experiment setup. Next, in Section 4.2, we compare and analyze the results obtained. In Section 4.3, we conduct an ablation study to evaluate the effectiveness of several components in AP. Lastly, in Section 4.4, we conduct a substitution study on the proposed AP.

### 4.1 Experiment Setup

**(1) Experiment Details.** To demonstrate that AP can work well with different pruning methods, we shortlist two classical and competitive pruning methods. The details are summarized as follows.

1. Pruning ResNet-20 on the CIFAR-10 dataset using Global Magnitude with and without AP.

2. Pruning VGG-19 on the CIFAR-10 dataset using Global Taylor with and without AP.

3. Pruning DenseNet-40 (Huang et al., 2017) on CIFAR-100 using Layer-Adaptive Magnitude-based Pruning (LAMP) (Lee et al., 2020) with and without AP.

4. Pruning MobileNetV2 (Sandler et al., 2018) on the CIFAR-100 dataset using Lookahead Pruning (LAP) (Park et al., 2020) with and without AP.

5. Pruning ResNet-50 (He et al., 2016) on the ImageNet (i.e., ImageNet-1000) using Iterative Magnitude Pruning (IMP) (Frankle & Carbin, 2019) with and without AP.

6. Pruning Vision Transformer (ViT-B-16) on CIFAR-10 using IMP with and without AP.

We train the network using SGD with He initialization (He et al., 2015), momentum $= 0.9$ and a weight decay of $1\text{e-}4$ (same as (Renda et al., 2019; Frankle & Carbin, 2019)). For the benchmark pruning method, we prune the network with a pruning rate $p = 20$ (i.e., 20% of existing weights are pruned) in 1 pruning cycle. After using AP, the overall pruning rate remains unchanged as 20%, but 2% of existing weights are pruned based on AP, while the other 18% of existing weights are pruned based on the benchmark pruning method to be compared with (see Algorithm 2). We repeat 25 pruning cycles in 1 run and use the early-stop top-1 test accuracy (i.e., the corresponding test accuracy when early stopping criteria for validation error is met) to evaluate the performance. The experimental results averaged over 5 runs and the corresponding standard deviation are summarized in Tables 1 - 6, where $\lambda$ is the percentage of weights remaining. The bolded results indicate that AP is significantly better than benchmarks results after accounting for the standard deviation.

**(2) Hyper-parameter Selection and Tuning.** To ensure fair comparison against prior results, we utilize standard implementations (i.e., network hyper-parameters and learning rate schedules) reported in the literature. Specifically, the implementations for Tables 1 - 6 are from (Frankle & Carbin, 2019), (Zhao et al., 2019), (Chin et al., 2020), (Renda et al., 2019) and (Dosovitskiy et al., 2020). The implementation details can be found in Section B.2 of the **Appendix**. In addition, we also tune hyper-parameters for each experiment setup mentioned above using the validation dataset via grid search. Some hyper-parameters are tuned as follows. (i) The training batch size is tuned from {64, 128, ...., 1024}. (ii) The learning rate is tuned from 1e-3 to 1e-1 via a stepsize of 2e-3. (iii) The number training epochs is tuned from 80 to 500 with a stepsize of 20. The validation performance using our tuned parameters are close to that of using standard implementations. Therefore, we use standard implementations reported in the literature to reproduce benchmark results.

| Original Top-1 Test Accuracy: 91.7% ($\lambda = 100\%$) | | | | |
|---|---|---|---|---|
| $\lambda$ | 32.8% | 26.2% | 13.4% | 5.72% |
| Global Magnitude | $90.3 \pm 0.4$ | $89.8 \pm 0.6$ | $88.2 \pm 0.7$ | $81.2 \pm 1.1$ |
| Global Magnitude with AP-Lite | $90.4 \pm 0.7$ | $90.2 \pm 0.8$ | $88.7 \pm 0.7$ | $82.4 \pm 1.4$ |
| Global Magnitude with AP-Pro | $90.7 \pm 0.6$ | $90.4 \pm 0.4$ | $89.3 \pm 0.8$ | $\mathbf{84.1} \pm \mathbf{1.1}$ |

Table 1: Performance (top-1 test accuracy ± standard deviation) of pruning ResNet-20 on CIFAR-10 using Global Magnitude with and without the proposed AP.

| Original Top-1 Test Accuracy: 92.2% ($\lambda = 100\%$) | | | | |
|---|---|---|---|---|
| $\lambda$ | 32.8% | 26.2% | 13.4% | 5.72% |
| Global Taylor | $90.2 \pm 0.5$ | $89.8 \pm 0.8$ | $89.2 \pm 0.8$ | $76.9 \pm 1.1$ |
| Global Taylor with AP-Lite | $90.5 \pm 0.8$ | $90.3 \pm 0.7$ | $89.7 \pm 0.9$ | $78.4 \pm 1.4$ |
| Global Taylor with AP-Pro | $90.8 \pm 0.6$ | $90.7 \pm 0.9$ | $90.4 \pm 0.8$ | $79.2 \pm 1.3$ |

Table 2: Performance (top-1 test accuracy ± standard deviation) of pruning VGG-19 on CIFAR-10 using Global Taylor with and without the proposed AP.

**(3) Reproducing Benchmark Results.** By using the implementations reported in the literature, we have correctly reproduced the benchmark results. For example, the benchmark results in our Tables 1 - 6 are comparable to Fig.11 and Fig.9 of (Blalock et al., 2020), Table.4 in (Liu et al., 2019), Fig.3 in (Chin et al., 2020), Fig. 10 in (Frankle et al., 2020), Table 5 in (Dosovitskiy et al., 2020), respectively.

**(4) Source Code & Devices:** We use Tesla V100 devices to conduct our experiments. The datasets are preprocessed using the conventional method. The source code is available at `https://github.com/Martin1937/Activate-While-Pruning`.

### 4.2 Performance Comparison

**(1) Performance using Classical Pruning Methods.** In Tables 1 & 2, we show the performance of AP using classical pruning methods (e.g., Global Magnitude, Global Taylor) via ResNet-20 and VGG-19 on CIFAR-10. We observe that as the percent of weights remaining, $\lambda$ decreases, the improvement of AP becomes larger. For example, in Table 1, the performance of AP-Lite at $\lambda = 26.2\%$ is 1.3% higher than the benchmark result. The improvement increases to 2.6% at $\lambda = 5.7\%$. Note that AP-Lite does not increase the algorithm complexity of existing methods. As expected, in Table 1, AP-Pro leads to a more significant improvement of 2.0% and 4.1% at $\lambda = 26.2\%$ and $\lambda = 5.7\%$, respectively. Similar performance trends can be observed in Table 2 as well. **The results for more values of $\lambda$ can be found in the Appendix**.

**(2) Performance using Competitive and Classical Pruning Methods.** In Tables 3 and 4, we show that AP can work well with Competitive pruning methods (e.g., LAMP, LAP). In Table 3, we show the performance of AP using LAMP via DenseNet-40 on CIFAR-100. We observe that AP-Lite improves the performance of LAMP by 1.2% at $\lambda = 13.4\%$ and the improvement increases to 1.6% at $\lambda = 5.7\%$. Note that AP-Lite does not increase the algorithm complexity of existing methods. For AP-Pro, it causes a larger improvement of 4.6% and 3.8% at $\lambda = 13.4\%$ and $\lambda = 5.7\%$, respectively. Similar performance trends can be observed in Table 4, where we show the performance of AP using LAP via MobileNetV2 on CIFAR-100.

**(3) Performance on ImageNet.** In Table 5, we show the performance of AP using Iterative Magnitude Pruning (IMP, i.e., the lottery ticket hypothesis pruning method) via ResNet-50 on ImageNet (i.e., the ILSVRC version) which contains over 1.2 million images from 1000 different classes. We observe that AP-Lite improves the performance of IMP by 1.5% at $\lambda = 5.7\%$. For AP-Pro, it improves the performance of IMP by 2.8% at $\lambda = 5.7\%$.

| Original Top-1 Test Accuracy: 74.6% ($\lambda = 100\%$) | | | | |
|---|---|---|---|---|
| $\lambda$ | 32.8% | 26.2% | 13.4% | 5.72% |
| LAMP | $71.5 \pm 0.7$ | $69.6 \pm 0.8$ | $65.8 \pm 0.9$ | $61.2 \pm 1.4$ |
| LAMP with AP-Lite | $71.9 \pm 0.8$ | $70.3 \pm 0.7$ | $66.6 \pm 0.7$ | $62.2 \pm 1.2$ |
| LAMP with AP-Pro | $72.2 \pm 0.7$ | $\mathbf{71.1 \pm 0.7}$ | $\mathbf{68.8 \pm 0.9}$ | $63.5 \pm 1.5$ |

Table 3: Performance (top-1 test accuracy $\pm$ standard deviation) of pruning DenseNet-40 on CIFAR-100 using Layer-Adaptive Magnitude Pruning (LAMP) with and without the proposed AP.

| Original Top-1 Test Accuracy: 73.7% ($\lambda = 100\%$) | | | | |
|---|---|---|---|---|
| $\lambda$ | 32.8% | 26.2% | 13.4% | 5.72% |
| LAP | $72.1 \pm 0.8$ | $70.5 \pm 0.9$ | $67.3 \pm 0.8$ | $64.8 \pm 1.5$ |
| LAP with AP-Lite | $72.5 \pm 0.9$ | $70.9 \pm 0.8$ | $68.2 \pm 1.2$ | $66.2 \pm 1.5$ |
| LAP with AP-Pro | $72.8 \pm 0.7$ | $71.4 \pm 0.8$ | $\mathbf{69.1 \pm 0.8}$ | $\mathbf{67.4 \pm 1.1}$ |

Table 4: Performance (top-1 test accuracy $\pm$ standard deviation) of pruning MobileNetV2 on CIFAR-100 using Lookahead Pruning (LAP) with and without the proposed AP.

| Original Top-1 Test Accuracy: 77.0% ($\lambda = 100\%$) | | | | |
|---|---|---|---|---|
| $\lambda$ | 32.8% | 26.2% | 13.4% | 5.72% |
| IMP | $76.8 \pm 0.2$ | $76.4 \pm 0.3$ | $75.2 \pm 0.4$ | $71.5 \pm 0.4$ |
| IMP with AP-Lite | $77.2 \pm 0.3$ | $76.9 \pm 0.4$ | $\mathbf{76.1 \pm 0.3}$ | $\mathbf{72.6 \pm 0.5}$ |
| IMP with AP-Pro | $77.5 \pm 0.4$ | $\mathbf{77.2 \pm 0.3}$ | $76.8 \pm 0.6$ | $\mathbf{73.5 \pm 0.4}$ |

Table 5: Performance (top-1 validation accuracy $\pm$ standard deviation) of pruning ResNet-50 on ImageNet using Iterative Magnitude Pruning (IMP) with and without AP.

| Original Top-1 Test Accuracy: 98.0% ($\lambda = 100\%$) | | | | |
|---|---|---|---|---|
| $\lambda$ | 32.8% | 26.2% | 13.4% | 5.72% |
| IMP | $97.3 \pm 0.6$ | $96.8 \pm 0.7$ | $88.1 \pm 0.9$ | $82.1 \pm 0.9$ |
| IMP with AP-Lite | $98.0 \pm 0.4$ | $97.3 \pm 0.7$ | $\mathbf{89.9 \pm 0.6}$ | $83.6 \pm 0.8$ |
| IMP with AP-Pro | $98.2 \pm 0.6$ | $97.6 \pm 0.5$ | $\mathbf{91.1 \pm 0.8}$ | $\mathbf{84.8 \pm 1.0}$ |

Table 6: Performance (top-1 test accuracy $\pm$ standard deviation) of pruning Vision Transformer (ViT-B-16) on CIFAR-10 using IMP with and without AP.

**(3) Performance on Competitive Networks (Vision Transformer).** Several recent works (Liu et al., 2021b; Yuan et al., 2021; Chen et al., 2021) demonstrated that transformer based networks tend to provide excellent performance in computer vision tasks. We now examine the performance of AP using Vision Transformer (i.e., ViT-B16 with a resolution of 384, pretrained on ImageNet dataset). We note that the ViT-B16 uses Gaussian Error Linear Units (GELU, GELU(x) = x$\Phi(x)$, where $\Phi(x)$ is the standard Gaussian cumulative distribution function) as the activation function. Similar to ReLU which blocks the negative pre-activation output, GELU heavily regularizes the negative pre-activation output by multiplying an extremely small value of $\Phi(x)$, suggesting that AP could be helpful with pruning GELU based models as well.

We repeat the same experiment setup as above and evaluate the performance of AP using ViT-B16 in Table 6. We observe that AP-Lite helps to improve the performance of IMP by 1.8% at $\lambda = 5.7\%$. For AP-Pro, it improves the performance of IMP by 3.3% at $\lambda = 5.7\%$.

| $\lambda$ | 32.8% | 26.2% | 13.4% | 5.72% |
|---|---|---|---|---|
| AP-Lite | **90.4 ± 0.7** | **90.2 ± 0.8** | **88.7 ± 0.7** | **82.4 ± 1.1** |
| AP-Lite-SOLO | 86.0 ± 1.0 | 84.3 ± 1.5 | 81.5 ± 2.0 | 74.5 ± 3.1 |
| AP-Lite-NO-WR | 87.5 ± 0.9 | 87.1 ± 1.2 | 84.7 ± 1.5 | 78.8 ± 2.3 |

Table 7: Ablation Study: Performance Comparison (top-1 test accuracy ± standard deviation) between AP-Lite and AP-SOLO, AP-Lite-NO-WR on pruning ResNet-20 on CIFAR-10 via Global Magnitude.

| $\lambda$ | 32.8% | 26.2% | 13.4% | 5.72% |
|---|---|---|---|---|
| AP-Pro | **90.8 ± 0.6** | **90.7 ± 0.9** | **90.4 ± 0.8** | **79.2 ± 1.3** |
| AP-Pro-SOLO | 85.8 ± 1.5 | 83.2 ± 1.7 | 81.5 ± 1.9 | 70.3 ± 2.7 |
| AP-Pro-NO-WR | 88.1 ± 1.2 | 86.3 ± 1.5 | 85.6 ± 1.5 | 74.8 ± 2.1 |

Table 8: Ablation Study: Performance Comparison (top-1 test accuracy ± standard deviation) between AP-Pro and AP-Pro-SOLO, AP-Pro-NO-WR on pruning VGG-19 using CIFAR-10 via Global Taylor.

| $\lambda$ | 32.8% | 26.2% | 13.4% | 5.72% |
|---|---|---|---|---|
| AP-Lite | **77.2 ± 0.3** | **76.9 ± 0.4** | **76.1 ± 0.3** | **72.6 ± 0.5** |
| AP-Lite-SOLO | 75.8 ± 0.5 | 74.3 ± 0.7 | 71.1 ± 0.6 | 68.5 ± 0.9 |
| AP-Lite-NO-WR | 76.3 ± 0.6 | 74.9 ± 0.8 | 73.2 ± 0.8 | 70.3 ± 1.1 |

Table 9: Ablation Study: Performance Comparison (top-1 test accuracy ± standard deviation) between AP-Lite and AP-Lite-SOLO, AP-Lite-NO-WR on pruning ResNet-50 on ImageNet via IMP.

### 4.3 Ablation Study

We now conduct an ablation study to evaluate the effectiveness of components in AP. Specifically, we remove one component at a time in AP and observe the impact on the pruning performance.

1. **AP-(Lite/Pro)-NO-WR**: Using AP without the weight rewinding step (i.e., remove step (ii) from Algo. 2). This aims to evaluate the effect of weight rewinding on the pruning performance.

2. **AP-(Lite/Pro)-SOLO**: Using only AP-(Lite/Pro) without the benchmark pruning method (i.e., in every pruning cycle, pruning weights only based on AP). This aims to evaluate if the pruning metric of AP alone can be used to evaluate the importance of weights.

In Tables 7 and 8, we conduct experiments of pruning ResNet-20 on the CIFAR-10 dataset using Global Magnitude (AP-Lite) and pruning VGG-19 on CIFAR-10 using Global Taylor (AP-Pro) respectively. In each case, we compare the performance of AP-(Lite/Pro)-NO-WR, AP-(Lite/Pro)-SOLO to AP-(Lite/Pro) so as to demonstrate the effectiveness of components in AP. We note that, same as before, we utilize the implementation reported in the literature. Specifically, the hyper-parameters and the learning rate schedule are from (Frankle & Carbin, 2019).

**Effect of Weight Rewinding.** In Tables 7 and 8, we compare the performance of AP-(Lite/Pro)-NO-WR to AP-(Lite/Pro). The key difference is that AP-(Lite/Pro) uses weight rewinding (see Algorithm 2) whereas the NO-WR approaches do not. We find that the performance of AP-(Lite/Pro) is always higher across all $\lambda$. For instance, at $\lambda = 5.72\%$, AP-Pro-NO-WR yields an accuracy of 74.8%, which is 4.4% lower than AP-Pro itself. This suggests the crucial role of weight rewinding in improving the performance.

| $\lambda$ | 32.8% | 26.2% | 13.4% | 5.72% |
|---|---|---|---|---|
| Global Magnitude (GM) | 90.3 (6.7%) | 89.8 (6.3%) | 88.2 (5.8%) | 81.2 (5.4%) |
| GM + AP-Pro | 90.7 (6.3%) | 90.4 (5.9%) | 89.3 (5.1%) | 84.1 (3.9%) |
| GM + AP-output | 90.6 (6.2%) | 90.5 (5.8%) | 89.4 (5.1%) | 84.4 (3.8%) |
| GM + AP-bias-0.2 | 90.1 (6.6%) | 89.4 (6.3%) | 88.1 (5.8%) | 82.3 (5.4%) |
| GM + AP-bias-0.5 | 89.9 (6.5%) | 89.6 (6.1%) | 86.9 (5.5%) | 82.8 (4.8%) |
| GM + AP-bias-1.0 | 89.5 (6.1%) | 88.7 (5.7%) | 87.0 (5.0%) | 81.5 (3.6%) |

Table 10: Performance comparison (i.e., top-1 test accuracy (dynamic DNR)) between AP-Pro and alternative methods in terms of dead neuron location and activation (rows 4 - 7) on ResNet-20 using CIFAR-10.

| $\lambda$ | 32.8% | 26.2% | 13.4% | 5.72% |
|---|---|---|---|---|
| LAP | 72.1 (7.8%) | 70.5 (6.9%) | 67.3 (6.5%) | 64.8 (6.1%) |
| LAP + AP-Pro | 72.8 (7.4%) | 71.4 (6.4%) | 69.1 (5.6%) | 67.4 (5.1%) |
| LAP + AP-output | 72.5 (7.6%) | 71.7 (6.2%) | 69.3 (5.8%) | 67.2 (5.3%) |
| LAP + AP-bias-0.2 | 71.8 (7.8%) | 70.2 (6.8%) | 66.8 (6.4%) | 63.3 (6.1%) |
| LAP + AP-bias-0.5 | 71.4 (7.7%) | 70.4 (6.6%) | 67.1 (6.2%) | 65.1 (5.8%) |
| LAP + AP-bias-1.0 | 71.2 (7.5%) | 70.1 (6.5%) | 67.7 (6.1%) | 65.9 (5.4%) |

Table 11: Performance comparison (i.e., top-1 test accuracy (dynamic DNR)) between AP-Pro and alternative methods in terms of dead neuron location and activation (rows 4 - 7) on MobileNetV2 using CIFAR-100.

**When AP Works Solely.** The pruning metric of AP (see Algorithm 1) aims to reduce dynamic DNR by pruning. We compare the performance of AP-(Lite/Pro)-SOLO to AP-(Lite/Pro) to evaluate if the pruning metric of AP can be used solely, without working with other pruning methods. In Tables 7 and 8, we observe that the SOLO methods perform much worse. For example, at $\lambda = 5.72\%$, the performance of AP-Lite-SOLO is 74.5, which is 7.9% lower than AP-Lite. It suggests that the pruning metric of AP alone is not suitable to evaluate the importance of weights. The effect of AP's metric on reducing dynamic DNR and its pruning rate $q$ on pruning performance are discussed in Section 5.

**More Results using ResNet-50 on ImageNet.** To further validate the results, we also conduct the ablation study using AP-Lite on ImageNet (ResNet-50). We summarize the results in Table 9, and we find that the results largely mirror those in Tables 7 and 8. Thus, both SOLO and NO-WR approaches perform significantly worse than the baseline.

### 4.4 Substitution Study

In this subsection, we conduct a substitution study on the proposed AP. Specifically, AP consists of two key components: dead ReLU neuron location and activation (see Algorithm 1). We replace one component at a time and observe the effect on dynamic DNR reduction and pruning performance (i.e., accuracy). we construct two variants of AP as follows and compare to the original AP-Pro.

1. **AP-output**: We replace the existing dead neuron location mechanism (i.e., weight movement) with directly observing the post-activated output of each ReLU neuron.

2. **AP-bias-k**: We replace the existing dead neuron activation mechanism (i.e., prune negative weights) by adding a constant value k to the bias of dead ReLU neurons.

The results of ResNet-20 & MobileNetV2 are summarized in Tables 10 - 11. The value in parentheses is dynamic DNR. Note that static DNR values remain roughly the same for all methods in each column and are therefore not shown. This is mainly because that majority of pruned weights are determined by the pruning method that AP works with (more details in Section 5). The findings from Tables 10 - 11 are as follows:

| AP Pruning Rate, $q$ | 1% | 2% | 3% | 5% |
|---|---|---|---|---|
| AP-Lite ($\lambda = 64.0\%$) | $89.8 \pm 0.1$ | $90.0 \pm 0.2$ | $89.5 \pm 0.4$ | $89.2 \pm 0.6$ |
| AP-Lite ($\lambda = 40.9\%$) | $88.2 \pm 0.4$ | $88.9 \pm 0.6$ | $88.5 \pm 0.7$ | $87.3 \pm 0.5$ |
| AP-Lite ($\lambda = 26.2\%$) | $87.1 \pm 0.5$ | $87.9 \pm 0.8$ | $86.7 \pm 0.8$ | $86.3 \pm 0.9$ |

Table 12: Performance (top-1 test accuracy $\pm$ standard deviation) of AP-Lite when iterative pruning ResNet-20 on CIFAR-10 with different pruning rate.

| AP Pruning Rate, $q$ | 1% | 2% | 3% | 5% |
|---|---|---|---|---|
| AP-Lite ($\lambda = 64.0\%$) | $91.1 \pm 0.2$ | $91.7 \pm 0.3$ | $90.2 \pm 0.5$ | $89.8 \pm 0.6$ |
| AP-Lite ($\lambda = 40.9\%$) | $88.7 \pm 0.5$ | $89.8 \pm 0.9$ | $89.3 \pm 1.1$ | $88.0 \pm 0.9$ |
| AP-Lite ($\lambda = 26.2\%$) | $87.9 \pm 0.8$ | $88.5 \pm 0.7$ | $88.1 \pm 1.3$ | $87.1 \pm 1.3$ |

Table 13: Performance (top-1 test accuracy $\pm$ standard deviation) of AP-Lite when iterative pruning VGG-19 on CIFAR-10 with different pruning rate.

1. **Effect of AP on reducing dynamic DNR.** When conventional pruning methods work with AP, the dynamic DNR is significantly reduced (compare GM to GM + AP-Pro in Table 10).

2. **Ceiling analysis on dead neuron location.** The performance of AP-Pro and AP-output is comparable in terms of reducing dynamic DNR and pruning performance. This suggests that AP-Pro works as expected and is able to locate dynamic dead neurons as if it directly observes the post-activated output.

3. **Implementation complexity.** We note that implementing AP-output is more complex than AP-Pro. AP-output requires practitioners to record the state of each neuron and then average over every training batch. For AP-Pro, we only need to compute the difference between two weight matrices. Given that AP-Pro can provide comparable performance to AP-output, and is relatively simpler to implement. As such, we still recommend using weight movements to locate dead ReLU neurons, as AP-Pro does now.

4. **Comparison to AP-bias-k.** When comparing AP-bias to AP-Pro, we find that a small value of k fails to reduce dynamic DNR as AP-Pro does. While for a large value of k, it can directly reduce dynamic DNR, but the performance performance is still not comparable to AP-Pro. We suspect that adding a large bias may hinder the optimization of the network during retraining, leading to uncompetitive results.

## 5 Reflections

In this section, we discuss several important points and present some experimental results.

**(1) Pruning Rate of AP, $q$.** Active Pruning removes $q\%$ of remaining parameters in every pruning cycle, so as to reduce dynamic DNR. The value of $q$ is usually much smaller than the pruning rate of the pruning method it works with. As an example, in Section 4, the overall pruning rate is fixed as 20% and 2% of weights are pruned based on Active Pruning, which is much smaller than the pruning rate of the benchmark method compared with (i.e., 18%). Adjusting the value of $q$ is a trade-off between pruning less important weights and reducing dynamic DNR. A large $q$ value indicates preferential reduction of dynamic DNR, while a small $q$ value means preferential removal of less important weights.

We repeat the experiments of pruning ResNet-20 on CIFAR-10 using Global Magnitude and AP-Lite. We note that the overall pruning rate is fixed as 20% and the pruning rate of AP increases from 1% to 5%. Correspondingly, the pruning rate of Global Magnitude decreases from 19% to 15%. The experimental results are summarized in Table 12. We observe that as we increase the pruning rate of AP from 2%, the performance tends to decrease. Similar performance trends can be observed using VGG-19 on CIFAR-10 as well (see Table 13). The theoretical determination of the optimal value of $q$ is clearly worth deeper thought. Alternatively, $q$ can be thought of as a hyper-parameter and tuned via the validation dataset and let $q = 2$ could be a good choice as it provides promising results in various experiments.

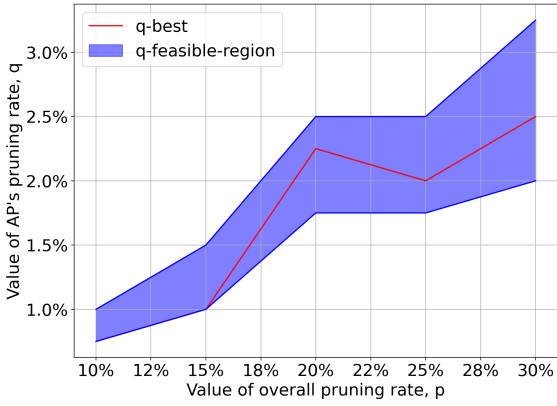 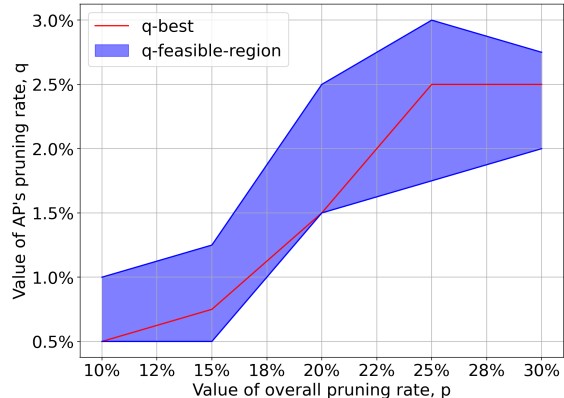

Figure 3: The value of $q$-best and $q$-feasible-region as $p$ gradually increases when iteratively pruning ResNet-20 (Left) and VGG-19 (Right) on CIFAR-10 using Global Magnitude and Global Taylor, respectively.

**(2) Relationship between p and q.** In the above experiments, we examine feasible values of AP's pruning rate $q$ when the overall pruning rate $p$ is fixed at 20%. We now conduct experiments to find feasible values of $q$ when $p$ is changing, so as to suggest the relationship between them. We conduct experiments with $p$ values from [10%, 15%, ..., 30%] on ResNet-20 and VGG-19. We gradually increase the value of $q$ with a step size of 0.25% and define two terms about $q$ as follows.

1. **q-best**: The value of $q$ that provides the best pruning performance in terms of accuracy.
2. **q-feasible-region**: The value region of $q$ that provides comparable performance to q-best (i.e., < 0.5% accuracy difference).

The results are depicted in Fig 3, where we make two observations: (i) As $p$ increases, $q$ should increase as well. (ii) A heuristic of $q = 0.1p$ seems to be a promising method to determine the value of $q$. Alternatively, $q$ can also be considered as a hyperparameter and tuned via validation dataset.

**(3) Comparison to Activation Sparsity Baselines.** We note that AP works in tandem with existing pruning methods and decreases the activation sparsity of pruned networks by pruning its negative weights. There could be other approaches which can decrease the activation sparsity as well, such as L1 regularization (Georgiadis, 2019) and boosted Hoyer Regularization (Kurtz & et al, 2020). Specifically, L1 regularization and boosted Hoyer regularization can be applied in the opposite direction to decrease the activation sparsity.

We now replace AP with activation sparsity baselines (e.g., boosted Hoyer regularization) and examine the performance when they work in tandem with existing pruning methods. The results are summarized in Tables 14 - 15. The takeaway message is two-fold: (i) When conventional pruning methods work with AP, this leads to a better performance than working with other activation sparsity baselines. The reason could be that AP explicitly targets dynamic dead neurons by pruning negative weights. This decreases the activation sparsity in a more precise manner. While other activation sparsity baselines use an augmented loss function and the decrease of activation sparsity becomes implicit and hard to control during optimization (i.e., the selection of which neuron is activated is not clear and there is no precise control). (ii) Interestingly, when we gradually prune the network, using conventional pruning methods with activation sparsity baselines outperforms the original counterpart (i.e., using conventional pruning method solely). For example, compare Global Magnitude + Booster Hoyer to Global Magnitude when $\lambda = 5.72\%$. This suggests that activation sparsity approaches can help to improve the performance of pruning methods. New methods to reduce activation sparsity to obtain more weighted sparsity are definitely worth exploring.

**(4) Static DNR.** During iterative pruning, the static DNR tends to increase as expected (see Fig. 1). It is interesting to mention that, after incorporating with AP, the static DNR of pruned networks remains almost the same as opposed to without. For example, the static DNR of Global Magnitude in Table 10 (second row) increases from 7.1% to 8.4% and finally reaches 15.9% when $\lambda$ decreases from 32.8% to 26.2% and finally to

| $\lambda$ | 32.8% | 26.2% | 13.4% | 5.72% |
|---|---|---|---|---|
| Global Magnitude (GM) | $90.3 \pm 0.4$ | $89.8 \pm 0.6$ | $88.2 \pm 0.7$ | $81.2 \pm 1.1$ |
| GM + AP-Pro | $90.7 \pm 0.6$ | $90.4 \pm 0.4$ | $89.3 \pm 0.8$ | $\mathbf{84.1 \pm 1.1}$ |
| GM + Booster Hoyer | $89.5 \pm 0.6$ | $88.7 \pm 0.8$ | $86.1 \pm 1.0$ | $82.5 \pm 0.7$ |
| GM + L1 Regularization | $89.8 \pm 0.5$ | $88.3 \pm 0.7$ | $85.6 \pm 0.9$ | $81.9 \pm 0.8$ |

Table 14: Performance (top-1 test accuracy ± standard deviation) of pruning ResNet-20 on CIFAR-10 using Global Magnitude with the proposed AP and other activation sparsity baselines.

| $\lambda$ | 32.8% | 26.2% | 13.4% | 5.72% |
|---|---|---|---|---|
| Global Taylor (GT) | $90.2 \pm 0.5$ | $89.8 \pm 0.8$ | $89.2 \pm 0.8$ | $76.9 \pm 1.1$ |
| GT + AP-Pro | $90.8 \pm 0.6$ | $90.7 \pm 0.9$ | $90.4 \pm 0.8$ | $\mathbf{79.2 \pm 1.3}$ |
| GT + Booster Hoyer | $89.8 \pm 0.7$ | $88.6 \pm 0.6$ | $87.1 \pm 1.2$ | $77.5 \pm 0.9$ |
| GT + L1 Regularization | $89.2 \pm 0.4$ | $88.1 \pm 0.7$ | $86.3 \pm 1.0$ | $77.1 \pm 1.2$ |

Table 15: Performance (top-1 test accuracy ± standard deviation) of pruning VGG-19 on CIFAR-10 using Global Taylor with the proposed AP and other activation sparsity baselines.

13.4%. After incorporating with AP-Pro (third row in Table 10), the static DNR almost remains the same. This is mainly because that the majority of pruned weights are still determined by the Global Magnitude (i.e., 18%) while only 2% of pruned weights are determined by AP.

The Theorem 1 shows that, in addition to dynamic DNR, reducing static DNR also can improve the upper bound of $I(X;T)$. In fact, reducing static DNR has been incorporated directly or indirectly into the existing pruning methods. As an example, LAMP (i.e., one Competitive pruning method used in performance evaluation, see Table 3) takes the number of unpruned weights of neurons/layers into account and avoids pruning weights from neurons/filters with less number of unpruned weights. This prevents neurons from being statically dead. Differing from existing methods, AP is the first method targeting the dynamic DNR. Hence, as a method that works in tandem with existing pruning methods, AP improves existing methods by filling in the gap in reducing dynamic DNR, leading to much better pruning performance.

**(5) Working with Non-ReLU based Networks.** We would like to highlight that AP also works well with non-ReLU based networks. For example, in Table 5, we show the performance of AP using Vision Transformer which uses GELU as the activation function. In this setup, AP also leads to an improvement of 2% - 3%. We posit that this could be due to the fact that, similar to ReLU which blocks the negative pre-activation output, GELU heavily regularizes the negative pre-activation output by multiplying an extremely small value of $\Phi(x)$, suggesting that AP could be helpful with pruning GELU based models as well.

**(6) Future Research.** (i) We only examine the effect of AP on network pruning using image datasets. In fact, AP may not be limited to this, but can also be applied to dynamic sparse training algorithms or for NLP tasks. (ii) The activation sparsity methods which enforce activation sparsity could also be used in the opposite way to decrease the activation sparsity. Such methods could be an alternative for AP and new methods to reduce activation sparsity to obtain more weight sparsity are definitely worth exploring. (iii) The pruning rate of AP $q$ is a important hyperparameter to tune and may significantly affect the performance. In above, we suggest the heuristic of $q = 0.1p$ to determine $q$ from the overall pruning rate $p$. A theoretical way to determine the value of $q$ is also worth exploring and $q$ may change in a non-linear way with $p$.

## 6 Conclusion

In this paper, we propose a new pruning method called Activate-while-Pruning (AP). Unlike existing pruning methods which remove less important parameters, the proposed AP works in tandem with existing pruning methods and aims to improve their pruning performance by de-sparsifying pruned networks. It is also

interesting to mention that the proposed AP studies the interaction of weight and activation sparsity from a new perspective, i.e., how to tradeoff activation sparsity for more weight sparsity.

Theoretically, we show the benefits of de-sparsifying pruned networks from the perspective of information bottleneck. Empirically, we use six different sets of experiments to demonstrate that AP can work well with a diverse range of networks (e.g., ResNet, VGG, DenseNet, MobileNet) and pruning methods (e.g., IMP, LAP, LAMP, etc) on both CIFAR-10/100 and ImageNet. It should be noted that AP is a generic approach, and by using the proposed AP, the pruning performance of existing pruning methods can be improved by 3% - 8%. Furthermore, we conduct an ablation study to further investigate and demonstrate the effectiveness of several key components that make up the proposed AP. Lastly, we conduct a substitution study to replace certain components in AP with alternative methods, further verifying the design of the proposed AP.

## Acknowledgements

This research is supported by A*STAR, CISCO Systems (USA) Pte. Ltd and the National University of Singapore under its Cisco-NUS Accelerated Digital Economy Corporate Laboratory (Award I21001E0002). Additionally, we would like to thank the members of the Kent-Ridge AI research group at the National University of Singapore for helpful feedback and interesting discussions on this work.

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
