# OpenReview forum: "AP: Selective Activation for De-sparsifying Pruned Networks"
_TMLR — Accepted by TMLR_

### Review · Reviewer_4xQt · 2023-06-03

**Summary Of Contributions:**

The paper proposes a mechanism (which can presumably be combined with other pruning algorithms) to improve the performance of the pruned ReLU network. The idea is to maximize the utilization rate of the neurons (of the pruned model) by pruning the model in a way that prevents outputting many zeros in the intermediate layer (i.e., minimizing the activation sparsity). To achieve this goal, authors propose a simple method: prune the negative weights, so that the pre-activation becomes larger, so that ReLU gets activated more often. The proposed method boost the performance of conventional pruning algorithms.

**Audience:**

Yes

**Broader Impact Concerns:**

This work may not really need a broader impact section.

**Claims And Evidence:**

Yes

**Requested Changes:**

- Please re-write the main text with more details about the existing works on activation sparsity.

- Compare with the activation sparsity baselines. For example, the one that uses Hoyer regularization can be used the other direction to decrease the activation sparsity, which could be a nice baseline method that this paper currently misses. The work also uses a specially designed activation function, and perhaps comparing with a similarly designed activation could help us better understand the benefit of the approach that modifies the weight pruning procedure.

- Please clarify the setups on Section 3.1.

- Please include the ImageNet1k experiment.

**Strengths And Weaknesses:**

__Strength.__ The proposed algorithm is quite well-motivated. The observations in Figure 1(right) is quite interesting (and makes a lot of sense), and could potentially inspire many future research efforts. I have not seen many results that explicitly study the intersection of the weight and activation sparsity (with a delightful exception of https://arxiv.org/abs/2112.13896 )---there could be some fundamental tradeoff between two notions of sparsity.

Another strength is the originality of the idea that authors propose to enhance the activation sparsity of the pruned models. As far as I know, typical approach to control the activation sparsity is by an explicit regularization of the intermediate activations. In contrast, this paper pioneers an alternative way, which is to perturb the weights along a specific direction (which makes most sense in the pruning context) so that the pre-activations shift toward more frequent activation. Perhaps a similar idea could also be used whenever a less frequent activation sparsity is needed.

__Weakness.__ The most significant weakness of this manuscript is that it almost completely disregards any related work on the activation sparsity. This paper revolves around a core concept, which the authors call "dead neuron rate," and this is actually a well-studied notion under the name of "activation sparsity" or sometimes ephemeral sparsity (see https://proceedings.mlr.press/v119/kurtz20a.html for instance). Hopefully to the authors, there is a small difference that usually the activation sparsity literature aims to maximize the activation sparsity, while this work aims to minimize. But still, the concept itself is not new, and there are some well-established techniques to control the activation sparsity which can generalize to the case of minimization. Also, the relationship between two notions have already been studied in some prior works (e.g., https://arxiv.org/abs/2112.13896 ).

Also, it seems like the manuscript is still under preparation. In the bottom of page 10, authors mention that "We intend to show results on larger datasets (e.g., ImageNet-1000) in the camera ready version." TMLR is a journal, where there is no fixed submission deadline. In this respect, deferring the ImageNet-1k result for the camera ready is not really understandable. If authors need more time, perhaps consider submitting the paper after all results are ready.

The writing in section 3.1. is somewhat unclear and can be improved. Could the readers clarify which pruning algorithm the authors are using? Are we doing as Han et al. (2015) does, or are we doing the full weight rewind as Frankle & Carbin (2019) does, or do a partial rewinding with the learning rate rewinding as Renda et al. does?

---

> ### Author Response · Authors · 2023-06-26
> **Author Responses to Reviewer 4xQt (1/2)**
>
> Dear Reviewer,
>
>
> Many thanks for your valuable time and effort in reviewing our manuscript. Special thanks for mentioning the activation sparsity papers. We are deeply impressed by your constructive comments. Below, we outline the revision made to the manuscript, in response to your comments and suggestions. **We note that all clarifications/changes below will be updated in the revised version**.
>
> **(1) [Existing works on activation sparsity]** We have carefully re-conducted the literature review and understood that our defined DNR is closely related to the activation sparsity. In many scenarios, they could be interchangeable. We will update the literature review and other relevant parts in the main paper to avoid confusion.
>
> Existing works [1, 2] on activation sparsity aim to boost network inference by enforcing activation sparsity. Very often, when the network is overfitting, enforcing activation sparsity may even bring some performance gain on classification accuracy. This is similar to network pruning, which introduces weight sparsity and may have some additional benefits, such as less memory cost.
>
> As you have mentioned, we note that our work links weight sparsity and activation sparsity from a new perspective, and provides the following insight: During iterative pruning, a heavily pruned network tends to be underfitting and fails to recover its performance through retraining. In such a case, slightly decreasing the activation sparsity using the proposed AP can help to further prune the network without the loss of accuracy, i.e., tradeoff activation sparsity for more weight sparsity. In summary, the proposed AP provides an approach to gain more weight sparsity with the cost of decreasing activation sparsity. This is different from existing works [3, 4], which aims to reduce computational cost by considering both activation and weight sparsity.
>
>
> **(2) [Comparison to activation baselines]** We note that AP works in tandem with existing pruning methods and decreases the activation sparsity of pruned networks by pruning its negative weights. As suggested, there could be other approaches which can decrease the activation sparsity as well, such as L1-regularization [2], Hoyer regularization [5] as mentioned by the reviewer. Specifically, the L1-regularizaition and Hoyer regularization can be used in the opposite direction to decrease the activation sparsity.
>
> We now replace AP with activation baselines (e.g., boosted Hoyer regularization [1]) and examine the performance when they work in tandem with existing pruning methods. The results are summarized in the following link. Please note that, the pruning method of Global Gradient is renamed as Global Taylor in the following link.
>
> https://drive.google.com/file/d/1Oz_ld4MjiC7kZzkTp-2Cqnz5js64SIYi/view?usp=sharing
>
> The takeaway message is two-fold: (i) When conventional pruning methods work with AP, this leads to a better performance than working with other activation sparsity baselines. The reason could be that AP explicitly targets dynamic dead neurons by pruning negative weights. This decreases the activation sparsity in a more precise manner. While other activation sparsity baselines use an augmented loss function and the decrease of activation sparsity becomes implicit and hard to control during optimization (i.e., The selection of which neuron is activated is not clear and there is no precise control). (ii) Interestingly, when we gradually prune the network, using conventional pruning method with activation sparsity baselines outperforms the original counterpart (i.e., using conventional pruning method solely). For example, compare global magnitude + Booster Hoyer to global magnitude when $\lambda$ = 5.72. This suggests that activation sparsity approaches can help to improve the performance of pruning methods. New methods which reduce activation sparsity to obtain more weighted sparsity are definitely worth exploring.

---

> > ### Author Response · Authors · 2023-06-26
> > **Author Responses to Reviewer 4xQt (2/2)**
> >
> > **(3) [Clarifications of setup on Section 3.1]** We are sorry about the incomplete description about the setup. In Section 3.1, we use global magnitude sa as Han et al (2015) with LR rewinding (Renda et al (2021)), but without weight rewinding (Frankle & Carbin (2019)). We will clarify this in the revised version.
> >
> > **(4) [ImageNet-1K Results on Ablation Study]** We note that we actually have provided ImageNet-1K results in the performance comparison (see Section 4.3 and Table 5). The ImageNet-1K results that we intended to show are for ablation study (Section 4.4). We have finished the simulation and summarized our new results on ImageNet-1K for ablation study in the following table.
> >
> > ______________________________________
> > $\lambda$     $~~~~~~~~~~~~~~~~~~~~~~~~~~~~~~~~~$           32.8%          $~~~~~~~~~~~~~~~~~$            26.2%      $~~~~~~~~~~~~~~~~~$              13.4%    $~~~~~~~~~~~~~~~~~$              5.72%
> > ______________________________________
> > AP-Lite           $~~~~~~~~~~~~~~~~~~~~~$     77.2 $\pm$ 0.3      $~~~~~~~~~~~$  76.9 $\pm$ 0.4     $~~~~~~~~~~~$   76.1 $\pm$ 0.3   $~~~~~~~~~~~$   72.6 $\pm$ 0.5
> >
> > AP-Lite-SOLO     $~~~~~~~~~~~$ 75.8 $\pm$ 0.5     $~~~~~~~~~~~$   74.3 $\pm$ 0.7     $~~~~~~~~~~~$   71.1 $\pm$ 0.6  $~~~~~~~~~~~$   68.5 $\pm$ 0.9
> >
> > AP-Lite-NO-WR  $~~~~~~~~$ 76.3 $\pm$ 0.6    $~~~~~~~~~~~$    74.9 $\pm$ 0.8    $~~~~~~~~~~~$    73.2 $\pm$ 0.8   $~~~~~~~~~~~$   70.3 $\pm$ 1.1
> > ______________________________________
> >
> > We note that the results reported above largely mirror those in Table 7 & 8, reaffirming the mechanism of the proposed AP.
> >
> >
> > $\newline$
> >
> > Thanks again for your valuable comments. We hope the changes mentioned above address your concerns. Please feel free to add if you have any new doubts/comments.
> >
> > $\newline$
> >
> > [1] Mark Kurtz, et al. Inducing and Exploiting Activation Sparsity for Fast Inference on Deep Neural Networks, ICML, 2020.
> >
> > [2] Georgios Georgiadis, et al. Accelerating Convolutional Neural Networks via Activation Map Compression, CVPR, 2019.
> >
> > [3] Tzu Hsien Yang, et al. Sparse ReRAM engine: joint exploration of activation and weight sparsity in compressed neural networks, 2019.
> >
> > [4] Kelvin Lee Hunter, et al. Two Sparsities Are Better Than One: Unlocking the Performance Benefits of Sparse-Sparse Networks, 2021.
> >
> > [5] Hoyer, P. O, et al. Non-negative matrix factorization with sparseness constraints. JMLR, 2004

---

### Review · Reviewer_YnV4 · 2023-06-21

**Summary Of Contributions:**

This paper studies dead neurons in pruned networks and classifies them as (1) neurons that have connections but give zero activations  neurons and (2) neurons without incoming connections. They observe that during pruning "static dead neuron" rate increases, however "dynamic dead neuron rate" (DNR) decreases. Authors argue that it would be better for learnability to have smaller DNR, thus they propose removing some of the negative valued incoming connections to push activations to the positive side of the ReLU. Authors combine this strategy with existing competitive pruning methods and prune a fraction of the weights (2%) using proposed approach while pruning the rest using the original pruning metric. Authors show improved generalization across different datasets and networks. Despite its strengths (see below), I found few things about the experimental evaluation concerning. Happy to update my review if these concerns are addressed.

Also, I had limited time to review this paper, please let me know if you have any questions.

**Audience:**

Yes

**Claims And Evidence:**

No

**Requested Changes:**

# Major
- *AP-Lite and AP-Pro* The difference between the two approach seems whether Iterative magnitude pruning is used. As argued by the authors AP can be applied to any pruning method and defining two version for 2 different pruning method seems unnecessary. As stated in (1a) and in the paper #training_steps is a hyper parameter increasing which often leads to better results [2].
- It is not clear why authors look at the movement of the neurons instead of their activations directly? I would guess with large enough batch size looking at the activations one can directly predict dead neurons. Without this it is not clear whether proposed pruning metric targets dynamic dead neurons directly. Similarly, effect of removing negative weights can be achieved by adding a constant value to the bias of the dead neurons. Would that work as well?
- What happens to Static Dead neurons in Table 11/12? It would be nice to add those numbers. To me it seems like proposed method increased static dead neurons by pruning weights from about to die neurons. If so it would be nice to say that.
- Is the hyper parameter search as explained in Section 4.1 done for both baseline pruning methods and AP separately? It is not clear hyper-parameters for which experiments are optimized.

# Minor
- I think we shouldn't be training VGG-19 on Cifar-10 and more importantly use it in pruning baselines. It is extremely over-parameterized and has much worse scaling than more recent architectures.
- I wouldn't call algorithms given SOTA. "Competitive" would be a better word. There are more recent works which report improved performance [3]. Global magnitude pruning often doesn't perform as well [4] gradual magnitude pruning (Zhu and Gupta).
- Better to call"Global Gradient"  as "Global Taylor" [5].
- There are quite a few references for lottery ticket hypothesis and IMP, which is not a competitive pruning method (see Renda et. al.) If authors like to keep these references, I would recommend adding more recent work discussing the limitations of lottery tickets [6].
- I don't think ViT paper as Cifar-10 pretraining. I assume authors prune the imagenet pretrained networks. Is that the case? If so, would be nice to mention this. Also in general when standard deviations overlap, one would conclude none of the methods are better, thus bolding both. It would be nice to do that in all results.
- "the the" -> "the"
- I would recommend authors to cite relevant work which looks into dead neurons in neural networks [7]

[2] https://arxiv.org/abs/1911.11134
[3] https://arxiv.org/abs/2106.12379
[4] https://arxiv.org/abs/2304.14082
[5] https://arxiv.org/abs/1906.10771
[6] https://arxiv.org/abs/2010.03533, https://arxiv.org/abs/2210.03044
[7] https://arxiv.org/abs/2302.12902, https://arxiv.org/abs/2108.06325

**Strengths And Weaknesses:**

# Strengths
- Paper is mostly well written and easy to follow
- Authors use different network architectures and datasets in image classification, which includes compact architectures like MobileNet and more recent Vision transformers with GELU activations.
- Authors investigate an interesting and under-studied area of neural network pruning

# Weaknesses
(1) I found few things about the experimental evaluation concerning. Happy to update my review if these concerns are addressed.

(a) It is not clear whether proposed algorithm AP uses more training steps than the baselines. Training the networks longer would give better pruning results, therefore baseline pruning methods should be trained longer using either Cosine cyclic learning rate (i.e. learning rate restarts) or regular linear scaling as used in [1].

(b) It is not clear whether proposed technic works as concluded (i.e. by reducing DNR) due to lack of relevant ablations (see requested changes below). It's not obvious why authors doesn't use

[1] https://arxiv.org/abs/1902.09574

---

> ### Author Response · Authors · 2023-06-28
> **Author Response to Reviewer YnV4 (1/3)**
>
> Dear Reviewer,
>
> We are sincerely thankful for your valuable time, prompt review and detailed comments. Special thanks for mentioning several interesting works. In the following, we clarify several points and outline changes to be made in the revised version, in response to your insightful suggestions.
>
>
> $\newline$
>
> **(1) [Difference between AP-Lite and AP-Pro]** The core of AP consists of the following three steps:
>
>    (i) Pruning the network according to AP’s pruning metric (not IMP’s pruning metric, see more details Algorithm 1).
>
>    (ii) Weight Rewinding
>
>    (iii) Retraining the network
>
> The difference between AP-Lite and AP-pro is that AP-Lite does the above mentioned three steps only once and AP-Pro repeats the above mentioned three steps in every pruning cycle. Also, we note that AP and IMP (iterative magnitude pruning) are using different pruning metrics. IMP prunes weights with the smallest magnitude to obtain weight sparsity while AP prunes negative weights of dead ReLU neurons with the aim of activating dead ReLU neurons. It is also interesting to mention that AP does not work alone and it works in tandem with existing pruning methods (e.g., IMP) and helps to de-sparsify pruned networks, leading to a better pruning performance.
>
> $\newline$
>
> **(2) [Effect of Longer training steps]** We note that the performance gain of AP is not from the longer training steps. Specifically,
>
> (i) AP-Lite uses the same training steps as the selected baseline. To clarify it, we show the algorithm of AP-Lite and select baselines (e.g., global magnitude) in the following link.
>
> https://drive.google.com/file/d/1RkrbuGnkxctNuMzqAR2BxKbYozf2S0-4/view?usp=sharing
>
> In Algorithm 4 & 5, we demonstrate the algorithm without and with AP-Lite. It can be seen that AP-Lite only has two additional steps (pruning & weight rewinding (blue color text), the retraining step actually overlaps with the pruning method (i.e., pruning method X) that AP works with), which are not related to training, but AP-Lite leads to a performance improvement of 2% - 3%. This suggests that the performance gain of the AP-Lite is not from the extra training steps.
>
> (ii) It is true that AP-Pro generally has longer training steps than selected baselines, but we would like to note that, for fair comparison, all selected benchmarks are tuned to provide the best performance, regardless of the number of training steps (see Section 4.1). It is also worth mentioning that we use the standard implementation (including training steps) reported in the literature and reproduce the comparable benchmark results to the literature (see Section 4.1, part (2) and (3)).
>
> Furthermore, we also re-conduct experiments in Section 4.2 and enable selected baselines with even longer training steps. The results are almost the same as what we have reported in Section 4.2
>
> (iii) As for the finding reported in the RigL paper, i.e., longer training steps lead to better performance, one possible reason could be that the authors are actually reporting the best testing accuracy while we are reporting the top-1 early-stop testing accuracy.
>
> We have carefully looked through the open source code of RigL paper in https://github.com/varun19299/rigl-reproducibility and do not see any early stopping criterion. The authors of RigL may record the testing accuracy in every training epoch and shortlist the best testing accuracy over all training epochs. In such a case, naturally, longer training steps may lead to better performance. While for us, we are actually reporting the top-1 early-stop testing accuracy (i.e., the testing accuracy when early stopping criteria is met).

---

> > ### Author Response · Authors · 2023-06-28
> > **Author Response to Reviewer YnV4 (2/3)**
> >
> > **(3) [Alternative methods for dead neuron location and activation]** In Algorithm 1, we summarize that AP consists of two components: (i) dead ReLU neuron location by observing the weight movement and (ii) dead ReLU neuron activation by pruning its negative weights. In fact, we have shown the effect of AP on reducing dynamic DNR in Table 11 & 12.
> >
> > Alternatively, as the reviewer has mentioned, dead ReLU neurons can also be located by directly checking the post-activated output. Similarly, dead neurons could be activated by adding a constant value to the bias of dead neurons. We now conduct new experiments to examine the effect of each component mentioned above in AP. Specifically, we replace one component at a time and construct several variants of AP as follows:
> >
> > (i) AP-output: We replace the existing dead neuron location mechanism (i.e., weight movement) with observing the post-activated output of each neuron, as suggested by the reviewer.
> >
> > (ii) AP-bias-k: We replace the existing dead neuron activation mechanism (i.e., prune negative weights) by adding a constant value k to the bias of dead neurons.
> >
> > We compare the above-mentioned two variants to the original AP (called AP-org)  and evaluate the performance from two perspectives: (i) the performance of reducing dynamic DNR. (ii) pruning performance. The new experimental results on ResNet-20 and MobileNetV2 are summarized in the following link.
> >
> > https://drive.google.com/file/d/1ThrTckF3-kpFB_FlHbGEois2CReFApxB/view?usp=sharing
> >
> > The takeaway message is three fold: (i) The performance of AP-org and AP-output is comparable in terms of reducing dynamic DNR and pruning performance (i.e., the accuracy of pruned networks). This suggests that AP-org works as expected and is able to locate dynamic dead neurons as if it directly observes the post-activated output. (ii) On the other hand, we note that implementing AP-output is more complex than AP-org. AP-output requires practitioners to record the state of every neuron and then average over every training batch. For AP-org, we only need to compute the difference between two weight matrices. Given that AP-org can provide comparable performance to AP-output, but is relatively simple to implement. We still recommend using weight movements to locate dead neurons, as AP-org does now. (iii) When comparing AP-bias to AP-org, we find that a small value of k fails to reduce dynamic DNR as AP-org does. While for a large value of k, it can directly reduce dynamic DNR, but the pruning performance is still not comparable to AP-org. We suspect that adding a large bias may hinder the optimization of the network during retraining, leading to uncompetitive results.
> >
> > **(4) [Static Dead Neurons in Tables 11 & 12]** We have augmented Tables 11 & 12 with static DNR in the following link
> >
> > https://drive.google.com/file/d/1d4C44jzst2n6HXMVOGGqQ7wkmma9h0jz/view?usp=sharing
> >
> > We note that for the pruning methods with and without AP, they have similar static DNR. This is mainly because that the majority of pruned weights are still determined by the pruning method that AP works with. For example, in Table 12, in each pruning cycle, global magnitude prunes 18% weights while AP only prunes 2% weights. Furthermore, AP only targets the dynamic DNR and has no effect on static DNR.
> >
> > **(5) [Parameter search in section 4.1]** Yes, the parameter search described in Section 4.1 is conducted for both baseline methods and AP. For each experiment, we redo the parameter search again. We find that the performance of our self-tuned hyper parameters are close to the standard implementation reported in the literature. Therefore, we use the standard implementation reported in the literature to reproduce the benchmark results (see Section 4.1, point (2))

---

> > > ### Author Response · Authors · 2023-06-28
> > > **Author Response to Reviewer YnV4 (3/3)**
> > >
> > > **(6) [Performance of VGG]** The main purpose of using VGG is to compare AP to the benchmark performance, as VGG is frequently used in many works. We understand that VGG is extremely overparameterized, and we are considering to shift the performance of VGG to the Appendix.
> > >
> > > **(7) [Literature Update]** Thanks for the references, we will include all papers you have mentioned into the updated literature. Also, we will re-conduct the literature review to make sure it is up-to-date.
> > >
> > > **(8) [Vit results and results reporting]** Yes, we are using the ImageNet pretrained model. We will also bold results whose standard deviation overlap with the best performance, as suggested.
> > >
> > > **(9) [Writing and Typos]** Thanks for your suggestions, we will correct typos and replace ‘SOTA’ and ‘Global Gradient’ with ‘competitive’ and ‘Global Taylor’, respectively.
> > >
> > > Thanks again for your valuable time. We hope the above-mentioned changes address your concerns and all changes will be updated in the revised version. Please feel free to add if you have any new doubts/comments.

---

### Review · Reviewer_jwS7 · 2023-06-23

**Summary Of Contributions:**

This paper introduces a novel method, AP,  for improving the performance of pruned neural networks which works in parallel with existing pruning methods. AP allocates a portion of the desired overall pruning ratio to negative weights which remain close to their initialized values at convergence. These weights are pruned in addition to those determined by the base method saliency criterion as required to meet the desired sparsity. AP reduces the DNR by increasing the number of active ReLU neurons, on average, across a given training dataset distribution. The author’s provide theoretical justification for their claims by considering how DNR relates to the Information Bottleneck framework. Empirical evidence of AP’s performance is presented across a number of datasets and network architectures for image classification.

**Audience:**

Yes

**Claims And Evidence:**

Yes

**Requested Changes:**

**Typos** (Must change for acceptance)
1. Formatting of in-text citations seems to be missing parentheses throughout the paper.
1. (“ii) AP-Pro which introduces an **addition** retraining step to the existing methods in every pruning cycle, but significantly improves the performance of existing methods.” -> Addition should be “additional” here.
1. “According to **Han et al. (2015), Han et al.** trained the unpruned network” -> Double in-text citation for Han et al. Only cite once.
1. A citation for VGG-19 appears to be missing. This should be added at the first mention of the network, similar to other network architectures.
1. “Haoran You and et al.” is listed as the author of “Max-affine spline insights into deep network pruning”. This should simply say “Haoran You et al.”. The corresponding in-text citation also shows the extra “and”.
1. Section 5.3 -> “**Differ** from existing methods, AP is the first…”. Should read: “Differing from …”

**Clarifications** (Further information will help secure acceptance, but is not specifically required)
1. “The success of ReLU is mainly **due to fact _(sic)_** that existing networks tend to be overparameterized and ReLU can easily regularize overparameterized networks” -> This claim does not appear to be supported by Glorot et al. (2011). The cited paper refers to several advantages related to vanishing / exploding gradient, sparse representations, and computational simplicity, but regularization is not listed as a motivation. Consider rephrasing this sentence or clarifying with additional works cited.
1. “For example, Liu et al. (2019a) demonstrated that training from scratch on the right sparse architecture yields better results than pruning from pre-trained models.” -> Please check this citation. The paper cited is for DARTS which is an AutoML technique that does not discuss sparsity as far as I am aware. It appears that this citation should be for Liu et al. (2019b). In any case, this is a strong claim to draw from Liu et al. (2019b) and is at odds with the majority of sparsity literature to my knowledge. Especially since Liu et al. explicitly noted that their claims only hold true for structured sparsity and they were unable to maintain accuracy with unstructured sparse networks on large scale datasets (imagenet). Given that the context of the paper under review is unstructured pruning, I encourage the authors to remove this claim unless better supporting justification can be provided.
1. In Fig 1 (Left), the figure is missing the 64% weight remaining column. What was the reason for this exclusion? All other weights remaining appear to follow the 20% pruning ratio. Consider including 64% or clarifying why this result is excluded.
1. While the pruning results augmented with AP do demonstrate improved generalization performance, in some cases this improvement is not clearly significant across the full range of sparsities considered. For the sake of clarity, I would encourage the authors to only bold tabulated results that are significant after accounting for the standard deviation. For instance, in Table 1 where lambda=32.8%, AP-Pro is bolded but the vanilla global magnitude (GM) results are not significantly worse considering the standard deviations of each result. I.e., 90.3+0.4 > 90.7-0.6.
1. Were any data augmentation schemes used in training? If so, those should be included in the experimental details section or in the appendix.
1. What initialization method was used to train the networks? This information should be included for reproducibility.
1. Table 7: Why do lambda=32.8% results for AP-Lite not match Table 1 for the same lambda? It appears that these should be identical runs?
1. For tables 7 and 8, why are the weights remaining depicted >> those shown in tables 1-4? As a reader, I am most interested in seeing how the ablations differ from the benchmark and results presented previously. Especially since AP is most beneficial in the highly sparse regime.

**Recommendations** (Changes not required, mere suggestions to consider)
1. The numerator and denominator in equation 1 could be expanded to clarify the contributions of static and dynamic DNR.  I.e., # of dead ReLU neurons + # of neurons with pruned incoming weights for the numerator. For denominator, consider clarifying that these are all "relu" neurons in the unpruned network (i.e., no conv, batch norm, etc. neurons included in this sum)
1. Figure 2 -> I suggest revising the AP % labels to depict (p-q)+q. As currently depicted, it appears to be p-q*q. The color labels help here but it could be more clear. Another recommendation here is to use the length of the color bars to show the overall model sparsity decreasing with each pruning cycle. I.e., Pruning Method X could show blue shading for 20% of the total bar at cycle 1, 20% + 16% for cycle 2, etc. This may help clarify to the reader how each pruning cycle removes a smaller and smaller portion of the total dense model weights.
1. In Section 4.1.2, the authors discuss a grid search conducted on a variety of hyperparameters. However, the results of this search do not appear to have been included as the “validation performance of using our tuned parameters are close to that of using standard implementations”. While I can appreciate the effort, I do not recommend including this description if the results are not included in the manuscript or appendices. Further, given that the purpose of this work is to establish the benefit of AP over existing methodologies, I believe the standard implementations already provide the most useful information to the reader when determining the overall impact of AP.
1. In Section 5.1, the authors discuss the effect of the pruning rate of AP, q. This is a very important inclusion as the introduction of any new hyperparameter deserves scrutiny. It would be interesting to see how this optimal q value changes with various p values. Perhaps q = 0.1 *p is a good heuristic across a multitude of p values?
1. “In this section, we first conduct experiments to evaluate the DNR during iterative pruning in Section 3.1.” -> I suggest referring to the current section only once. I.e., “In section 3.1, we first …”
1. The results section is needlessly verbose in some paragraphs with several sentences simply repeating the results listed in tabular form elsewhere. Where these sentences are used to emphasize a finding they work well; however, each paragraph in section 4.2 contains a similarly formatted sentence which diminishes the emphasizing effect. I encourage the authors to simply remove sentences that repeat the tabulated results without expanding or providing further relevant information.
1. Further investigation of AP by applying it to dynamic sparse training algorithms (SET, RigL, etc.) would be of additional interest to the broader sparse neural network research community.
1. Only image classification tasks were considered. It would be interesting to see how AP performs on NLP tasks as well.




**Strengths And Weaknesses:**

**Strengths**
1. As far as I am aware, the proposed method is novel in its approach and provides interesting insights into the relationship between parameter sparsity and ephemeral sparsity induced by commonly used nonlinear activation functions.
1. Extensive experiments are conducted across a wide variety of datasets and network architectures.
1. This paper studies a very timely and important topic as sparse neural networks are an important ingredient in improving the overall efficiency of serving ML models at scale.
1. The proposed method complements existing pruning algorithms and appears to demonstrate moderate improvements across each dataset/architecture pair studied. Of particular note are improvements in generalization performance at high sparsities.
1. The Information Bottleneck analysis provides a compelling theoretical justification for the proposed method and generalization improvements observed.

**Weaknesses**
1. The heuristic used to locate dead ReLU neurons could also be used to locate neurons that were initialized at values close to their final converged value. I am curious to know if the authors conducted any empirical studies to determine how accurate this heuristic is in practice at locating truly dead neurons.
1. A more formal analysis of the additional FLOPs and/or wall-clock timings required to perform the AP-Pro algorithm would help the reader understand what kind of overhead should be expected when adopting this method.
1. The paper could use another proofread as there were some minor typos and formatting issues that must be addressed. See the Requested Changes section below for more details.
1. Presentation of results could be improved by combining tables and using more plots, where appropriate. Further, maintaining the same sparsities across Tables 1-8 would make the ablation studies more compelling.
1. Additional investigations into applying AP in the context of dynamic sparse training and NLP tasks would be of additional interest to the broader sparse neural network research community.

---

> ### Author Response · Authors · 2023-06-30
> **Author Responses to Reviewer jwS7 (1/2)**
>
> Dear Reviewer,
>
> We are greatly thankful for your valuable time and effort in reviewing our manuscript. We highly appreciate your detailed comments and suggestions. We have carefully considered your suggestions and outline changes to be made in the revised version as follows.
>
> **[Typos]**
>
> We are sorry about the careless mistakes. We will correct all typos and formatting issues as you have suggested.
>
> **[Clarifications]**
>
> (1) Yes, we will rephrase the sentence and add more references as follows. We will cite three additional works [1, 2, 3], where [1] demonstrates there is a significant redundancy in deep learning models, and [2, 3] suggests that ReLU enables network with sparsity, leading to various properties, such as less sensitive to noise, better generalization, etc.
>
>
> (2) We will remove the claim about Liu et al (2019b) paper as suggested.
>
> (3) The main reason for the exclusion of the 64% weight remaining column is that its dynamic DNR does not reduce much from 80% remaining weights (i.e., from 7.7% to 7.58%), which is less obvious to observe the change in dynamic DNR, but we do see your point and will put it back to avoid confusion.
>
> (4) We will bold results as suggested.
>
> (5) Yes, we used the conventional augmentation schemes during training, which included random crop and image flips. These details will be clarified, and the code will be released at the camera ready stage.
>
> (6) We use the kaiming uniform [4] (also known as He initialization) for weight initialization.
>
> (7) Yes, the results of AP-Lite in Table 7 should match those in Table 1. We have carefully checked our code and found that we forgot to use LR rewinding [5] when generating results for Table 7. To avoid confusion, we have rerun the experiments of Table 7 with LR rewinding and updated its performance in the following link.
>
> https://drive.google.com/file/d/16M6xhR_0I02uaqdkxs3xFdrNtK6G6XVT/view?usp=sharing
>
> (8) In Tables 7 & 8, initially, we feel the takeaway message is clear (i.e., weight rewinding in the proposed AP is important and AP cannot work alone) when we show results up to $\lambda$ = 32.8%. After reading your comment, we decide to augment Tables 7 & 8 with the results of more sparse pruned networks, so as to better convey the takeaway message. Please see our latest results in the following link.
>
> https://drive.google.com/file/d/16M6xhR_0I02uaqdkxs3xFdrNtK6G6XVT/view?usp=sharing

---

> > ### Author Response · Authors · 2023-06-30
> > **Author Responses to Reviewer jwS7 (2/2)**
> >
> > **[Recommendations]**
> >
> > (1) We will expand equation 1 as suggested.
> >
> > (2) We will update Figure 2 as suggested.
> >
> > (3) We will update our description in Section 4.1.2 as suggested.
> >
> > (4) **[Relationship between p and q]** As suggested, we conduct new experiments to examine the relationship between the overall pruning rate, **p** and the pruning rate of AP, **q**. Note that **(p - q)%** is pruned by the pruning method that AP works with.
> >
> > We conduct experiments with p values from [10%, 15%, …, 30%] on ResNet-20 and VGG-19. We gradually increase the value of **q** with a step size of 0.25% and define two terms relating to **q** as follows.
> >
> > (i) **q-best**: the value of **q** that gives the best pruning performance in terms of accuracy.
> >
> > (ii) **q-feasible-region**: the value region of **q** that provides comparable performance to **q-best** (<0.5% accuracy difference).
> >
> > We plot the trend of **q-best** and **q-feasible-region** vs **p** in the following link.
> >
> > https://drive.google.com/file/d/1Wie21QU-06bnTMTI0MPcOlTzgiy-H37O/view?usp=sharing
> >
> > The takeaway message is two-fold: (i) As **p** increases, **q** should increase as well. (ii) The heuristic of **q = 0.1 * p** seems to be a promising method to determine the value of **q**.
> >
> > (5) We will rewrite the sentence as suggested.
> >
> > (6) We will rewrite Section 4.2 as suggested.
> >
> > (7 & 8) Honestly speaking, it is a bit difficult for us to confidently report the effect of the proposed AP on dynamic sparse training algorithms and NLP tasks, within such a short period of time. We do see your points about the potential applications of AP and will highlight this in future research, which will hopefully inspire more interesting works.
> >
> > $\newline$
> >
> > **[Additional Experiments to Examine if AP can accurately locate dead neurons]**
> >
> > We note that the proposed AP consists of two components: (i) locating dead ReLU neurons via weight movement and (ii) activating dead ReLU neurons by pruning. In fact, we have shown the effect of AP on reducing dynamic DNR in Table 11 & 12. The results demonstrate the proposed AP can effectively reduce the dynamic DNR, suggesting that AP is able to locate dead ReLU neurons.
> >
> > To further examine it, we conduct a substitution study to replace the weight movement mechanism with directly observing the post-activated ReLU output. This leads to a new variant of AP called AP-output. We now compare the proposed AP-Pro to AP-output in terms of reducing dynamic DNR and pruning performance (i.e., accuracy). The results using ResNet-20 and MobileNetV2 are summarized in the following link.
> >
> > https://drive.google.com/file/d/1NSTC3HZv4uMyf1GooIOeemZUNAgvnvje/view?usp=sharing
> >
> > The takeaway message from the above table is two-fold: (i) The performance of AP-Pro and AP-output is comparable in terms of reducing dynamic DNR and pruning performance (i.e., the accuracy of pruned networks). This suggests that AP-Pro works as expected and is able to accurately locate dynamic dead neurons as if it directly observes the post-activated output. (ii) On the other hand, we note that implementing AP-output is more complex than AP-Pro. AP-output requires practitioners to record the state of every neuron and then average over every training batch. For AP-Pro, we only need to compute the difference between two weight matrices. Given that AP-Pro can provide comparable performance to AP-output, but is relatively simple to implement. We still recommend using weight movements to locate dead neurons, as AP-org does now.
> >
> > We believe the new experiments mentioned above could better verify the effectiveness of AP on locating dead neurons and we will create a new subsection called substitution study in the revised paper.
> >
> > $\newline$
> >
> > Thanks again for your valuable time. Please feel free to add if you have any new doubts/comments.
> >
> > $\newline$
> >
> > [1] Misha Denil, et al. Predicting parameters in deep learning, 2014
> >
> > [2] Yifei Wang, et al. Overparameterized ReLU neural networks learning the simplest model: neural isometry and phase transition, 2023.
> >
> > [3] Zonglin Li, et al. The lazy neuron phenomenon: On emergence of activation sparsity in transformers, 2023.
> >
> > [4] Kaiming He, et al. Delving deep into rectifiers: Surpassing human level performance on ImageNet classification, 2015.
> >
> > [5] Alex Renda, et al. Comparing Rewinding and Fine-tuning in Neural Network Pruning, 2020

---

> > > ### Comment · Reviewer_jwS7 · 2023-07-03
> > > **Response to Authors' rebuttal**
> > >
> > > I thank the authors for their detailed rebuttal comments. I am especially impressed by the authors' inclusion of additional experiments to analyze the relationship between p and q in more detail and to further demonstrate how their heuristic for locating dead neurons appears to closely match the performance of finding the dead neurons explicitly in the new AP-output method. I understand that inclusion of experiments on DST methods and NLP tasks is not possible within the constrained review timeline. However, I appreciate the nod to these research directions in the future work section.
> > >
> > > In general, the vast majority of my concerns have been addressed by the changes discussed above. A handful of minor concerns remain pending:
> > > * “For example, (Han et al., 2015) trained the unpruned network” -> The citation here should be the in-text version (ie., `\citet{}` command in latex to produce ‘For example, Han et al. (2015) trained…’)
> > > * The citation for VGG should be in parenthesis (i.e, VGG (Simonyan & Zisserman, 2014)...
> > > * In my opinion, there is still some potential to improve the presentation of results as per Weakness 4 and Recommendation 6 of my original review. I encourage the authors to consider further improvements to the presentation for their camera-ready submission.
> > >
> > > The remaining above-noted concerns are minor and I expect they will be revised in the camera-ready version. No further revisions by the authors are required at this time to address my review comments / concerns. Based on the above discussion and revisions, I am pleased to recommend acceptance of this work.

---

> > > > ### Author Response · Authors · 2023-07-04
> > > > **Author Response to Reviewer jwS7**
> > > >
> > > > Dear Reviewer,
> > > >
> > > > Thanks again! We highly appreciate your valuable time and great attention to details. We are glad to hear your positive feedback and will further update the camera ready version based on your suggestions.

---

### Author Response · Authors · 2023-07-02
**New Revision Uploaded based on Reviewers' Suggestions**

Dear Action Editor and Reviewers,

On behalf of all the contributing authors, I would like to express our sincere appreciation of your professional work and reviewers' constructive and insightful comments. Based on your suggestions, we have revised the manuscript extensively and added new experimental results to better convey the ideas of the whole paper. We have uploaded a new version of our manuscript and summarized the changes as follows:

**[General Changes]**

1. Corrected typos and formatting issues.
2. Updated the literature review with more citations and discussions on dead ReLU neurons and activation sparsity.
3. Clarified experiment setups, certain paragraphs have been rewritten, and some figures have been updated to avoid confusions.
4. Updated the bold texts in the tabulated results.

**[Specific Changes]**

**To Reviewer 4xQt:**

1. We have added the comparison to activation sparsity baselines in Section 5 (see Tables 14 & 15).

2. We have included the ImageNet results for ablation studies (see Table 9).


**To Reviewer YnV4:**

1. We have added a new subsection called substitution study to examine the effect of certain replacements as suggested, so as to demonstrate if AP can accurately locate and activate dead ReLU neurons (see Section 4.4, Table 10 & 11).

2. We have described the results of static DNR in Section 4.4 and Section 5.


**To Reviewer jwS7:**

1. We have included new experiments to examine the relationship between the overall pruning rate p and AP's pruning rate q (Section 5 and Figure 3).

2. We have corrected the results in Table 7.

3. We have included results for more sparse networks (up to $\lambda$ = 5.72\%) in Tables 7, 8 and 9.

4. We have added a new subsection called substitution study to examine the effect of certain replacements as suggested, so as to demonstrate if AP can accurately locate and activate dead ReLU neurons (see Section 4.4, Tables 10 & 11).


We hope these changes can address reviewers' suggestions and concerns. Please feel free to add if you have any new comments or doubts. Thanks again for your valuable time.

---

### Decision · Action_Editors · 2023-08-07

**Recommendation:** Accept as is

**Comment:**

The submission clearly meets the bar in terms of claims and evidence and interest to TMLR's audience. I encourage the authors to incorporate the following reviewer feedback in the camera-ready version:
* As per Reviewer YnV4's suggestion, using the `<X>+AP` notation, where X is either "One-Shot" or "IMP".
* Incorporating a discussion on the effect (or lack thereof) of training length in the main text.
* Combining tables and using more plots, where appropriate, and maintaining the same sparsities across Tables 1-8.
* Removing sentences that repeat the tabulated results without expanding or providing further relevant information.



**Audience:**

All reviewers agree that the submission is of interest to the neural network pruning research community and also to a broader proportion of the TMLR audience:
* "[...] publication of this paper will be a valuable contribution to pruning literature." (Reviewer jwS7)
* "provides interesting insights into the relationship between parameter sparsity and ephemeral sparsity induced by commonly used nonlinear activation functions." (Reviewer jwS7)
* "[...] investigate an interesting and under-studied area of neural network pruning." (Reviewer YnV4)
* "[...] this paper could definitely add interesting ideas to the activation sparsity literature." (Reviewer 4xQt)
* "observations [are] quite interesting (and [make] a lot of sense), and could potentially inspire many future research efforts." (Reviewer 4xQt)

**Claims And Evidence:**

Reviewers shared initial concerns that were ultimately addressed to their satisfaction. Their consensus is that the submission's claims are supported by accurate, convincing, and clear evidence.